

# What is an archaeon and are the Archaea really unique?

Ajith Harish

Department of Cell and Molecular Biology, Program in Molecular Biology, Uppsala University, Uppsala, Sweden

## ABSTRACT

The recognition of the group Archaea as a major branch of the tree of life (ToL) prompted a new view of the evolution of biodiversity. The genomic representation of archaeal biodiversity has since significantly increased. In addition, advances in phylogenetic modeling of multi-locus datasets have resolved many recalcitrant branches of the ToL. Despite the technical advances and an expanded taxonomic representation, two important aspects of the origins and evolution of the Archaea remain controversial, even as we celebrate the 40th anniversary of the monumental discovery. These issues concern (i) the uniqueness (monophyly) of the Archaea, and (ii) the evolutionary relationships of the Archaea to the Bacteria and the Eukarya; both of these are relevant to the deep structure of the ToL. To explore the causes for this persistent ambiguity, I examine multiple datasets and different phylogenetic approaches that support contradicting conclusions. I find that the uncertainty is primarily due to a scarcity of information in standard datasets—universal core-genes datasets—to reliably resolve the conflicts. These conflicts can be resolved efficiently by comparing patterns of variation in the distribution of functional genomic signatures, which are less diffused unlike patterns of primary sequence variation. Relatively lower heterogeneity in distribution patterns minimizes uncertainties and supports statistically robust phylogenetic inferences, especially of the earliest divergences of life. This case study further highlights the limitations of primary sequence data in resolving difficult phylogenetic problems, and raises questions about evolutionary inferences drawn from the analyses of sequence alignments of a small set of core genes. In particular, the findings of this study corroborate the growing consensus that reversible substitution mutations may not be optimal phylogenetic markers for resolving early divergences in the ToL, nor for determining the polarity of evolutionary transitions across the ToL.

Corresponding author
Ajith Harish, ajith.harish@gmail.com

## INTRODUCTION

The recognition of the Archaea as the so-called "third form of life" was made possible in part by a new technology for sequence analysis, oligonucleotide cataloging, developed by Fredrik Sanger and colleagues in the 1960s (*Woese, 2004*; *Woese & Fox, 1977*). Carl Woese's insight of using this method, and the choice of the small subunit ribosomal

RNA (16S/SSU rRNA) as a phylogenetic marker, not only put microorganisms on a phylogenetic map (or tree), but also revolutionized the field of molecular systematics that Zuckerkandl and Pauling had previously alluded to (*Zuckerkandl & Pauling, 1965*). Comparative analysis of organism-specific oligonucleotide signatures in SSU rRNA led to the recognition of a distinct group of microorganisms (*Woese, 2004*). Initially referred to as Archaebacteria, SSU rRNAs in these unusual organisms had "sequence signatures" distinct from other Bacteria (Eubacteria), and they were later found to be different from those of eukaryotes (Eukarya) as well. Many other signatures, including molecular, biochemical as well as ecological features, corroborated the uniqueness of the Archaea. Thus the archaeal concept was established (*Woese, 2004*). Accordingly, the five-kingdoms of life classification scheme (*Whittaker, 1969*) was replaced by the three-domains of life classification (*Woese, Kandler & Wheelis, 1990*).

The genomic representation of microbial biodiversity, particularly of the Archaea, has since expanded significantly. This is largely due to advances in environmental genome sequencing: the sampling of microbial DNA directly from the environment without the need for culturing (*Rinke et al., 2013*; *Sunagawa et al., 2015*). Since large-scale exploration by the means of environmental genome sequencing became possible almost a decade ago, there has also been a palpable excitement and anticipation of the discovery of a fourth form of life or a "fourth domain" of life (*Wu et al., 2011*). The reference here is to a fourth form of cellular life, but not to viruses, which some have already proposed to be the fourth domain of the tree of life (ToL) (*Boyer et al., 2010*). If a fourth form of life were to be found, what would the distinguishing features be, and how could it be measured, defined and classified?

Rather than the discovery of a fourth domain, and contrary to the expectations, however, current discussion is centered around the return to a dichotomous classification of life (*Harish & Kurland, 2017a*; *Harish, Tunlid & Kurland, 2013*; *Williams et al., 2013*); despite hundreds of novel phyla descriptions (*Hug et al., 2016*; *Parks et al., 2017*). The proposed dichotomous classification schemes, however, are in sharp contrast to each other, depending on: (i) whether the Archaea constitute a monophyletic group—a unique line of descent that is distinct from those of the Bacteria as well as the Eukarya; and (ii) whether the Archaea form a sister clade to the Eukarya or to the Bacteria. Both the issues stem from difficulties involved in resolving the deep branches of the ToL (*Gouy, Baurain & Philippe, 2015*; *Harish, Tunlid & Kurland, 2013*; *Williams et al., 2013*).

The twin issues, first recognized based on discordant tree topologies derived from single-gene analyses (*Lake, 1986*; *Tourasse & Gouy, 1999*), continue to be the subjects of a long-standing debate, which remains unresolved despite large-scale analyses of multi-gene datasets (*Da Cunha et al., 2017*; *Rinke et al., 2013*; *Spang et al., 2015*; *Williams & Embley, 2014*; *Zaremba-Niedzwiedzka et al., 2017*). In addition to the choice of genes to be analyzed, the choice of the underlying character evolution-model is at the core of contradictory results that either support the three-domains tree in which the Archaea are monophyletic and sister to Eukarya (*Da Cunha et al., 2017*; *Rinke et al., 2013*; *Woese, Kandler & Wheelis, 1990*); or the Eocyte tree, in which the Archaea are paraphyletic, and some Archaea (Crenarchaea/eocytes) are sister to Eukarya

(*Lake et al., 1984*; *Williams & Embley, 2014*; *Zaremba-Niedzwiedzka et al., 2017*).
A third competing hypotheses is the two-empires tree, which places Archaea sister to
Bacteria (*Brinkmann & Philippe, 1999*; *Mayr, 1998*), but is rarely considered. In many cases
of such systematic discordances, adding more data, either as enhanced taxon sampling or
enhanced character sampling, or both, can resolve ambiguities (*Salichos & Rokas, 2013*;
*Zwickl & Hillis, 2002*). However, as the taxonomic diversity and evolutionary distance
increases among the taxa studied, the number of conserved marker genes that can be
aligned for phylogenetic analyses decreases. Accordingly, recovery of the historical
signal in multiple sequence alignments (MSAs) by phylogenetic analyses is restricted to a
small set of conserved loci or genes—50 at most (*Zaremba-Niedzwiedzka et al.,
2017*)—usually referred to as "universal core genes" (*Williams et al., 2013*; *Woese, 2002*).

Recovery of historical signal in MSA by tracing the history of single-residue
substitutions is the standard molecular phylogenetic approach. However, several
conserved genomic loci, that is, the loci per se, are a distinct class of phylogenetic markers
(*Hillis, 1999*; *Rokas & Holland, 2000*). Phylogenetic signal can be recovered from
covariation patterns among genomes of highly conserved loci such as introns, mobile
elements, protein-coding and non-coding genes, protein-domains, and other genomic
features (*Harish, Tunlid & Kurland, 2013*; *Hillis, 1999*; *Snel, Bork & Huynen, 1999*;
*Tarver et al., 2013*; *Wang & Caetano-Anolles, 2006*; *Yang, Doolittle & Bourne, 2005*).
Genomic features are underutilized in phylogenomic studies, even though their
advantages over single-residue substitutions, for instance, low observed levels of
homoplasy, has been known for long. This was, initially, due to the practical difficulties in
collecting multiple characters per clade, and over a broad range of taxonomic groups,
to assemble large-scale datasets that is necessary for statistically robust inferences
(*Hillis, 1999*). Although assembling large datasets is no longer a barrier for the estimation
of phylogeny using genomic features, it was, until recently, limited to parsimony methods
(*Harish, Tunlid & Kurland, 2013*; *Kim & Caetano-Anollés, 2011*). Analysis of datasets
with hundreds of taxa is now feasible using both maximum likelihood (ML) (*Fang et al.,
2013*) as well as Bayesian (*Harish & Kurland, 2017a*) methods, but the statistical
behavior and robustness to rate heterogeneities have not yet been carefully characterized.

Altogether, independent phylogenomic analyses that employ different, but overlapping
datasets yield alternative tree topologies with incompatible branching patterns
(*Da Cunha et al., 2017*; *Harish & Kurland, 2017a*; *Kim & Caetano-Anollés, 2011*;
*Rinke et al., 2013*; *Spang et al., 2015*; *Williams & Embley, 2014*). Contradicting conclusions
are also supported when different analytical approaches are applied to the same
datasets (*Da Cunha et al., 2017*; *Harish & Kurland, 2017a*; *Rinke et al., 2013*; *Spang et al.,
2015*; *Williams & Embley, 2014*). Despite the contradictions, the branches typically
receive high branch support values—statistical measures of confidence in a given branch—
and thus provide equivocal support for contradicting scenarios for the early diversification
of Archaea.

Here, to understand the source of such conflicting results, I examine different
phylogenomic datasets and alternative approaches used to resolve such conflicts.
Specifically, the quality of different types of molecular features, and the utility or
a lack thereof, of such data for resolving complex phylogenetic problems is assessed. I find that a primary cause for this persistent ambiguity is that the "information" necessary to resolve these conflicts is inadequate in the standard "universal core genes" datasets employed routinely to reconstruct the global ToL. In contrast, covariation patterns of unique genomic loci provide for sufficient information for a reliable resolution of the conflicts. Resolving the evolutionary relationships of Archaea to other taxa, however, depends on the placement of the root of the ToL (*Brinkmann & Philippe, 1999*; *Harish, Tunlid & Kurland, 2013*). Using an expanded taxonomic sampling of recently described groups of Archaea, including the TACK (Thaumarchaeota, Aigarchaeota, Crenarchaeota, and Korarchaeota), DPANN (Diapherotrites, Parvarchaeota, Aenigmarchaeota, Nanoarchaeota, Nanohaloarchaea), and Asgard Archaea (*Zaremba-Niedzwiedzka et al., 2017*), I re-evaluate the utility of directional evolution-models (*Harish & Kurland, 2017a*; *Klopfstein, Vilhelmsen & Ronquist, 2015*) to identify the root of the ToL. I find that the resolution of the phylogenetic radiations, in deep time, based on genomic features is robust against potential artifacts due to biases in character-specific and lineage-specific rate heterogeneity (heterotachy) as well as composition bias.

Accordingly, phylogenetic modeling of the evolution of genomic features validates the uniqueness (monophyly) of the Archaea, and the placement of Archaea sister to Bacteria (*Brinkmann & Philippe, 1999*; *Harish & Kurland, 2017b*; *Harish, Tunlid & Kurland, 2013*; *Mayr, 1998*). Further, the independent and parallel diversification of eukaryote and akaryote species is corroborated (*Forterre & Philippe, 1999*; *Harish & Kurland, 2017a*, *2017c*). Findings from this case study on Archaea are broadly applicable to the problem of incongruence that is often encountered in efforts to resolve certain other early divergences in the ToL, for example, at the root of the eukaryote-ToL (*Derelle et al., 2015*; *He et al., 2014*) and the metazoan-ToL (*Philippe et al., 2011*; *Shen, Hittinger & Rokas, 2017*; *Whelan et al., 2015*).

Importantly this study shows that, despite the presence of conflicting signals that arise from disparate processes of reticulate evolution, the earliest divergences of life can be reconstructed reliably using genomic signatures of evolutionary transitions. I discuss underutilized approaches to recover phylogenetic signal in genome sequence data that are valuable to minimize phylogenetic uncertainties. Finally, I discuss simple but important, yet undervalued, aspects of phylogenetic hypothesis testing, which together with the new approaches hold promise to resolve these long-standing issues effectively.

## DATA AND METHODS

### Data sources and data (character) types analyzed

Five datasets, one single-locus dataset and four multi-locus phylogenomic datasets were analyzed in this study (Table 1). All datasets, except one, were obtained from previous studies that focused on resolving the phylogenetic relatedness of Archaea to Eukarya and Bacteria. A new dataset was assembled for this study, to include recently discovered taxa (see details below). To distinguish the different character codings used to represent genomic loci in the data matrices, characters are classified as either (i) Elementary molecular characters: single-residue (nucleotide and amino acid) characters in MSA;
**Table 1 Phylogenomic datasets that use different character types, and the source of the datasets.**

| Dataset | Character type | Number of taxa | Number of loci | Number of characters |
|---|---|---|---|---|
| SSU rRNA[a] | Elementary (nucleotide) | 140 | 1 gene | 1,462 |
| Core-genes-I[b] | Elementary (amino acid) | 44 | 29 genes | 8,563 |
| Core-genes-II[a] | Elementary (amino acid) | 96 | 48 genes | 9,868 |
| SCOP-domains-I[c] | Complex (protein domain) | 141 | 1,732 domains | 1,732 |
| SCOP-domains-II[d] | Complex (protein domain) | 222 | 1,738 domains | 1,738 |

Notes:
SCOP, structural classification of proteins.
[a] *Zaremba-Niedzwiedzka et al. (2017)*.
[b] *Williams & Embley (2014)*.
[c] *Harish, Tunlid & Kurland (2013)*.
[d] Present study.

or, (ii) Complex molecular characters: genomic features that are distinct permutations of elementary characters. In this study, complex characters are genomic loci that correspond to protein-domains; specifically, domains that are identified from experimentally determined three-dimensional (3D) structures according to the structural classification of proteins (SCOP) scheme for identifying homologous domains (*Gough et al., 2001*; *Murzin et al., 1995*). A detailed description of the different datasets is as follows,

i) Elementary character datasets: MSA datasets were obtained *as-is* from previous studies (Table 1); a single-gene nucleotide MSA of the SSU rRNA and two amino acid MSAs of concatenated universal core genes. The universal core genes (henceforth simply core-genes) are conserved genes that are found in all organisms, which function in the transcription and translation processes of gene expression. Genes that are included in phylogenomic data matrices mainly encode components of the translation apparatus, ribosomal proteins, and translation factors as well as a few components of RNA polymerases. Different MSAs with overlapping sets of core-genes were obtained (Table 1): (a) Core-genes-I dataset is a MSA of 29 genes (*Williams & Embley, 2014*); (b) Core-genes-II dataset is a MSA of 48 genes (*Zaremba-Niedzwiedzka et al., 2017*). The number of core-genes sampled or the extent of overlap between different datasets depends on taxon sampling and the criteria applied for filtering data to be analyzed (*Williams & Embley, 2014*). For instance, different sequence similarity thresholds used to identify orthologs, or the level of stringency applied to the definition of universal markers: either to be present in every taxon sampled (universal) or to allow for gene absences to be coded as missing data (nearly universal). Together, these criteria determine the size of the data matrix in terms of the number of characters considered to be informative to test phylogenetic hypotheses (Table 1).

ii) Complex character datasets: homologous protein-domains were coded with non-arbitrary presence–absence state labels (*Lewis, 2001*). Data matrices of SCOP-domains were assembled from genome annotations available through the SUPERFAMILY HMM library and genome assignments server, v. 1.75 (http://supfam.org/SUPERFAMILY/) (*Gough et al., 2001*; *Oates et al., 2015*).
When genome annotations were unavailable from the SUPERFAMILY database, curated reference proteomes were obtained from the universal protein resource (http://www.uniprot.org/proteomes/). SCOP-domains were annotated using the Hidden Markov Model (HMM) library and genome annotation tools as recommended by the SUPERFAMILY resource. A more detailed description of the protocol can be found in *Harish, Tunlid & Kurland (2013)*. Two datasets (Table 1) with overlapping taxon samples were assembled as follows,

a) SCOP-I dataset: a 141-species dataset was obtained from a previous study (*Harish, Tunlid & Kurland, 2013*). The broadest possible taxonomic diversity of sequenced genomes available at the time was sampled. An equal number of species, 47 each, were sampled from Archaea, Bacteria, and Eukarya. The number of genomes was limited by the number of unique genera of Archaea for which genome sequences were available at the time of the study. 1,732 of the 2,000 distinct SCOP-domains are represented in this sampling.

b) SCOP-II dataset: the 141-species dataset was updated with representatives of novel species described recently, largely with archaeal species from the TACK group (*Guy & Ettema, 2011*), DPANN group (*Rinke et al., 2013*), and Asgard group including the Lokiarchaeota (*Zaremba-Niedzwiedzka et al., 2017*). In addition, species sampling was enhanced with representatives from the (unclassified) candidate phyla described for bacterial species (*Anantharaman et al., 2016*) and with unicellular species of eukaryotes, to a total of 222 species. 1,738 SCOP-domains are represented in this sampling. The complete list of the species with their respective Taxonomy IDs is available in Table S1. Unlike with curating MSA datasets, data (character) filtering is not required to assemble protein-domain datasets.

## Exploratory data analysis

Data-display networks (DDNs) were constructed with SplitsTree 4.1 (*Huson & Bryant, 2006*). Split networks were computed using the neighbor-net method from the observed genetic distances (*p*-distances) of the taxa for both nucleotide- and amino acid characters in the core-genes datasets. Split networks of the protein-domain characters were computed from Hamming distance. The network diagrams were drawn with the equal angle algorithm.

## Phylogenetic analyses

### Core-genes datasets

The best-fitting amino acid substitution model was chosen using Smart Model Selection (*Lefort, Longueville & Gascuel, 2017*) and ModelFinder (*Kalyaanamoorthy et al., 2017*). Both model selection tests chose the LG model of amino acid substitution (*Le & Gascuel, 2008*) to be the best-fitting model, for both core-genes datasets (Tables S2 and S3). However, analysis here was restricted to the core-genes-I dataset due to a relatively smaller taxon sampling (44 species) compared to the core-genes-II dataset (96 species), since the computational time required for estimating trees is significantly lesser. Moreover,

the general conclusions, including paraphyly of Archaea, based on these datasets are consistent (*Williams & Embley, 2014*; *Zaremba-Niedzwiedzka et al., 2017*). Extensive analyses of these two concatenated core-genes datasets are reported in the original studies.

Unrooted (undirected) trees were estimated with both the rate-homogeneous as well as rate-heterogeneous versions of the LG model implemented in PhyML 3.0 (*Guindon et al., 2010*). Character-specific rate heterogeneity (CSRH) was approximated using the discrete gamma distribution (*Yang & Roberts, 1995*) with four, eight, and 12 rate categories, LG+G4, LG+G8, and LG+G12, respectively. More complex models (Table S2) that account for invariable characters (LG+GX+I) and/or models that compute alignment-specific character-state frequencies (LG+GX+F) were also used, but the trees inferred were identical to trees estimated from LG+GX models, and therefore not reported here. Log likelihood ratio (LLR) was calculated as the difference in the raw log likelihood scores for each model.

### SCOP-domain datasets

Both unrooted (undirected) trees and intrinsically rooted (directed) trees were estimated. The Mk model (*Lewis, 2001*) applicable to complex features coded as binary-state characters is the most widely implemented model for phylogenetic inference in both ML and Bayesian phylogenetic methods. However, only reversible models are implemented in ML software at present. Both reversible and directional evolution-models as well as model selection routines are implemented in MrBayes 3.2 (*Klopfstein, Vilhelmsen & Ronquist, 2015*; *Ronquist et al., 2012*). Directional evolution refers to either non-reversibility of character transitions or non-stationarity of state frequencies, or both, along the tree. Since standard reversible models assume stationarity of character frequencies and reversibility of character transitions, the likelihood scores are independent of the placement of the root. Directional evolution-models, however, relax the standard assumptions to allow non-stationarity and non-reversibility of character transitions such that frequency of characters at the root of the tree is allowed to be different from the rest of tree (*Klopfstein, Vilhelmsen & Ronquist, 2015*). Therefore, likelihood scores depend on the placement of the root.

## Root inference

The placement of the root is crucial to determine the monophyly (or non-monophyly) of a taxonomic group as well as sister-group relationships. Several methods can be used to identify the root of phylogenetic trees: Paleontological (temporal) data, outgroup rooting, the molecular clock and directional evolution-models (*Huelsenbeck, Bollback & Levine, 2002*). The former two are not applicable to the global ToL as there are no known fossils or outgroups that can be employed. Directional models are able to identify the correct rooting of trees without the use of an outgroup or other prior knowledge (*Huelsenbeck, Bollback & Levine, 2002*; *Klopfstein, Vilhelmsen & Ronquist, 2015*). The utility and efficacy of the directional evolution-model, to detect non-stationarity and non-reversibility is rigorously characterized with simulations and empirical datasets in

previous studies (*Harish & Kurland, 2017a*; *Klopfstein, Vilhelmsen & Ronquist, 2015*). The utility of directional evolution-models to root the global ToL, and the suitability of models that are both non-stationary and non-reversible has been tested and discussed extensively in previous studies using the SCOP-I dataset (*Harish & Kurland, 2017a, 2017b*). In this study, the placement of the root is analyzed further using both non-clock and relaxed-clock models using the SCOP-II dataset (i.e., with an expanded taxonomic diversity).

## Robustness of root placement

Robustness of root placement against potential systematic biases with focus on errors due to CSRH as well as lineage-specific rate heterogeneity (LSRH or heterotachy) was analyzed in this study. Robustness of root placement against other potential errors was assessed extensively, and reported in four earlier studies (see Supplementary Methods). Briefly, these include impact of: (1) species (taxon) sampling, (2) inclusion and exclusion of lineage-specific domains (characters), (3) small (or large) genome-size bias, (4) uncertainty in domain assignments with HMM models (ascertainment bias), and (5) quality of genome sequence data/annotations.

In the present study, sensitivity of the directional model to CSRH was analyzed by varying the number of rate categories under the Gamma rate variation model. In addition, to test if the placement of the root is biased due to LSRH, relaxed-clock models implemented in MrBayes were used under non-stationarity. Relaxed-clock models allow rates to vary across lineages, in addition to rate variation across characters. Three different relaxed-clock models where the rate variation across lineages is modeled according to Compound Poisson process (CPP) model, Brownian motion model (TK02), and Independent Gamma Rate model, with default priors for branch lengths were used (for details see Supplementary Methods).

Altogether, 15 different models of increasing complexity that assume (1) rate homogeneity or different extents of CSRH, LSRH; (2) reversibility or non-reversibility, and (3) stationarity or non-stationarity of the evolutionary process were characterized, using the SCOP-II dataset. The model complexity is proportional to the number of assumptions incorporated in the model. In each case, two independent runs of Metropolis-coupled MCMC samplings were used with four chains each, sampling every 500th generation. MCMC sampling was run until convergence, unless mentioned otherwise. Convergence was assessed through the average standard deviation of split frequencies (ASDSF, <0.01) for tree topology and the potential scale reduction factor (PSRF = 1.00) for scalar parameters, unless mentioned otherwise. The first half of the generations was discarded as burn-in. Bayes factors for model comparison were calculated using the harmonic mean estimator in MrBayes. The log Bayes factor (LBF) was calculated as the difference in the marginal log likelihoods for each model.

Convergence between independent runs was generally slower for directional models compared to the reversible models. When convergence was extremely slow (requiring more than 100 million generations and/or more than 21 days run-time) topology constraints based on the clusters derived from the unrooted trees were applied to improve

convergence rates. As such these clusters/constraints corresponded to named taxonomic groups, for example, Fungi, Metazoa, Crenarchaeota, etc. Convergence assessment between independent runs was relaxed for three (out of 15) models that did not converge sufficiently at the time of submission: non-clock rooted trees corresponding to root-R2 (ASDSF 0.05; PSRF 1.04), root-R3 (ASDSF 0.02; PSRF 1.01) and relaxed-clock rooted tree using the CPP model (ASDSF 0.03; PSRF 1.05). In these three cases specified, the difference in bipartitions is in the shallow parts (minor branches) of the tree, but not within the deeper nodes (major branches). For assessing well-supported major branches of the tree, ASDSF values between 0.01 and 0.05 may be adequate, as recommended by the authors (*Ronquist, Huelsenbeck & Teslenko, 2011*).

## RESULTS

### Information in core-genes datasets is inadequate to resolve the archaeal radiation

Data-display networks are useful to examine and visualize character conflicts in phylogenetic datasets, especially in the absence of prior knowledge about the source of such conflicts (*Huson & Bryant, 2006*; *Morrison, 2009*). While congruent data will be displayed as a tree in a DDN, incongruences are displayed as reticulations in the tree. Figure 1A shows a neighbor-net analysis of the SSU rRNA alignment used to resolve the phylogenetic position of the recently discovered Asgard Archaea (*Zaremba-Niedzwiedzka et al., 2017*). The DDN is based on character distances calculated as the observed genetic distance (*p*-distance) of 1,462 characters, and shows the total amount of conflict in the dataset that is incongruent with character bipartitions (splits). The edge (branch) lengths in the DDN correspond to the support for the respective splits. Accordingly, two well-supported sets of splits for the Bacteria and the Eukarya are observed. The Archaea, however, does not form a distinct, well-resolved/ well-supported group, and is unlikely to correspond to a monophyletic group in a phylogenetic tree.

Likewise, the concatenated protein sequence alignment of the so-called "genealogy defining core of genes" (*Woese, 2002*)—a set of conserved single-copy genes—also does not support a unique archaeal lineage. Figure 1B is a DDN derived from a neighbor-net analysis of 8,563 characters in 29 concatenated core-genes (*Williams & Embley, 2014*), while those in Figs. 1C and 1D are based on 9,868 characters in 48 concatenated core-genes (also from *Zaremba-Niedzwiedzka et al., 2017*). However, in Fig. 1D, amino acids in the MSA are recoded as a reduced set of alphabets using the SR-4 (from 20 to 4) recoding scheme (*Susko & Roger, 2007*). Even taken together, none of the standard marker gene datasets are likely to support the monophyly of the Archaea—a key assertion of the three-domains hypothesis (*Woese, Kandler & Wheelis, 1990*). Simply put, there is not enough information in the core-genes datasets to resolve the archaeal radiation, or to determine whether the Archaea are really unique compared to the Bacteria and Eukarya. However, other complex features—including molecular, biochemical, and phenotypic characters, as well as ecological adaptations—support the uniqueness of the Archaea (*Garrett, 1985*; *Valentine, 2007*; *Woese, 2004*).

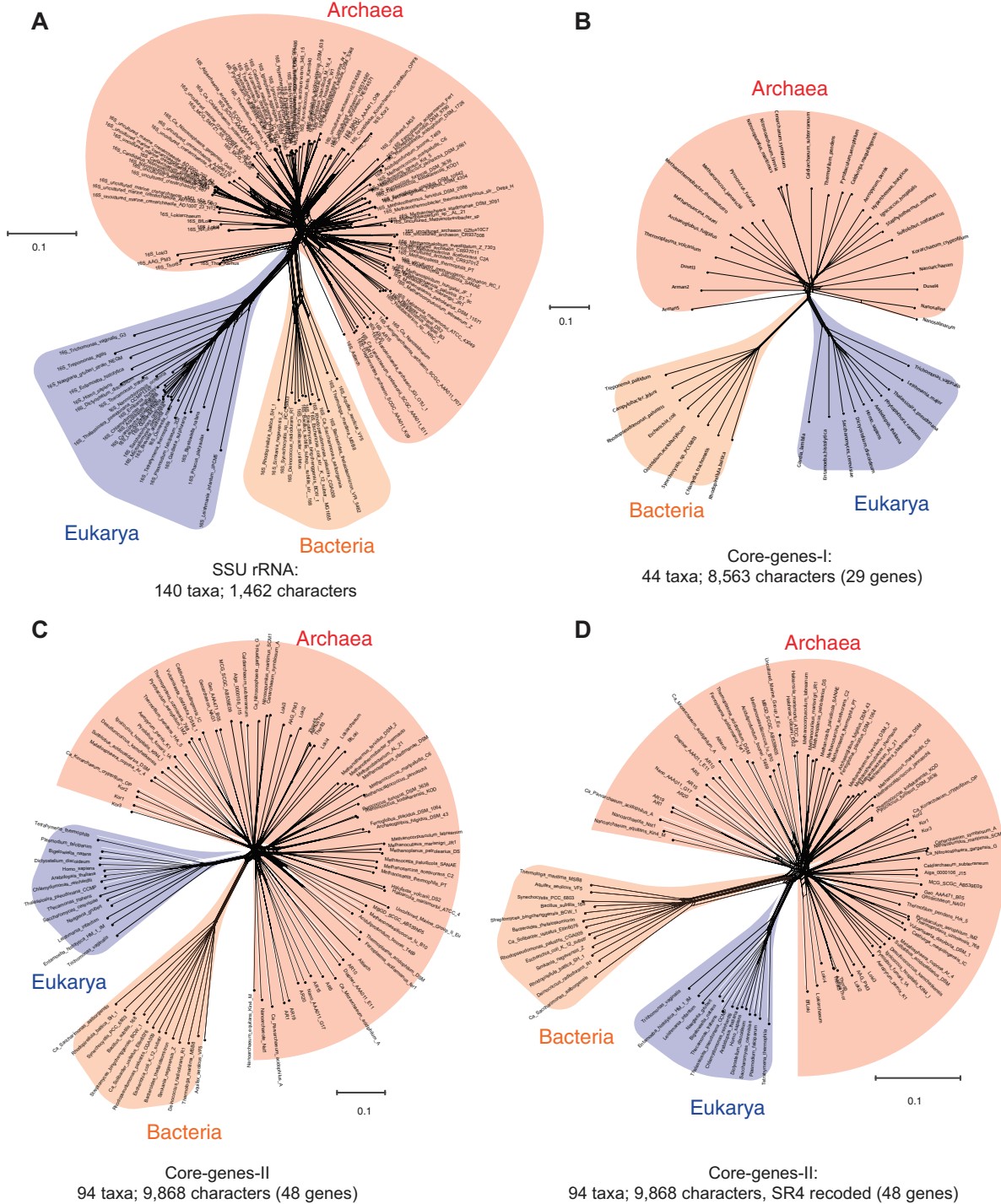

**Figure 1 Data-display networks (DDN) depicting the character conflicts in datasets that employ different character types: nucleotides or amino acids, to resolve the tree of life.** (A) SSU rRNA alignment of 1,462 characters. Concatenated amino acid sequence alignment of: (B) 29 genes, 8,563 characters (Core-genes-I dataset); (C) 48 genes, 9,868 characters (Core-genes-II dataset); and (D) SR4 recoded core-genes-II dataset (data simplified from 20 to four character-states). Each network is constructed from a neighbor-net analysis based on the observed genetic distance (*p*-distance) and displayed as an equal angle split network. Edge (branch) lengths correspond to the support for character bipartitions (splits), and reticulations in the tree correspond to character conflicts. The scale bar represents the split support for the edges. Conflicts in character partitions that are incongruent with a tree appear as reticulations in the DDN. Source of the datasets is as specified in Table 1.

## Complex molecular characters minimize uncertainties regarding the uniqueness of the Archaea

A nucleotide is the smallest possible locus, and an amino acid is a proxy for a locus of a nucleotide triplet. Unlike the elementary amino acid- or nucleotide-characters in the core-genes dataset (Fig. 1), the DDN in Fig. 2 is based on complex molecular characters: relatively larger genomic loci that are formed by distinct permutations of elementary characters. In this case the loci correspond to protein-domains, typically ~200 amino acids (600 nucleotides) long. Each protein-domain is unique: with a distinct sequence profile, 3D structure and function (Fig. 3). Neighbor-net analysis of protein-domain data coded as binary characters (presence–absence) is based on the Hamming distance (identical to the *p*-distance used in Fig. 1). Here, the Archaea also form a distinct well-supported cluster, as do the Bacteria and the Eukarya.

Figure 2A is a DDN based on the dataset that includes protein-domain cohorts of 141 species, used in a phylogenomic analysis to resolve the uncertainties at the root of the ToL (*Harish & Kurland, 2017a*). Compared to the data in Fig. 1, the taxonomic diversity sampled for the Bacteria and Eukarya is more extensive, but less extensive for the Archaea; it is composed of the traditional groups Euryarchaeota and Crenarchaeota. Figure 2B is a DDN of an enriched sampling of 81 additional species, which includes representatives of the newly described archaeal groups: TACK, DPANN, and Asgard (Lokiarchaeota, Thorarchaeota, Odinarchaeota, Heimdallarchaeota). In addition, species sampling was enhanced with representatives from the candidate phyla described for Bacteria, and with unicellular species of Eukarya. The complete list of species analyzed in Table S1.

Notably, the extension of the protein-domain cohort is insignificant, from 1,732 to 1,738 distinct domains (characters). Based on the well-supported splits in the DDN that form a distinct archaeal cluster, the Archaea are likely to be a monophyletic group (or a clade) in phylogenies inferred from these datasets.

## Employing complex molecular characters maximizes the representation of orthologous non-recombining genomic loci, and thus phylogenetic signal

Despite the superficial similarity of the DDNs in Figs. 1 and 2, they are both qualitatively and quantitatively different codings of genome sequences. As opposed to tracing the history of, at most 50 loci, in the standard core-genes datasets (Fig. 1), up to 30-fold more information (1,738 loci) is represented when genome sequences are coded as protein-domain characters (Fig. 2). Currently ~2,000 unique domains are described by SCOP (*Andreeva et al., 2014*). The phyletic distribution of 1,738 distinct domains identified in the 222 representative species sampled here is shown in a Venn diagram (Fig. 3C). 1,190 out of 1,738 domains (~70%) are shared widely such that 855 (~50%) are distributed across all the three major taxa and the rest shared between two of the three taxa.

A closer look at the core-genes datasets shows that the regions of the MSAs that are retained after data filtering correspond to the distinct protein-domains (Fig. 3A;
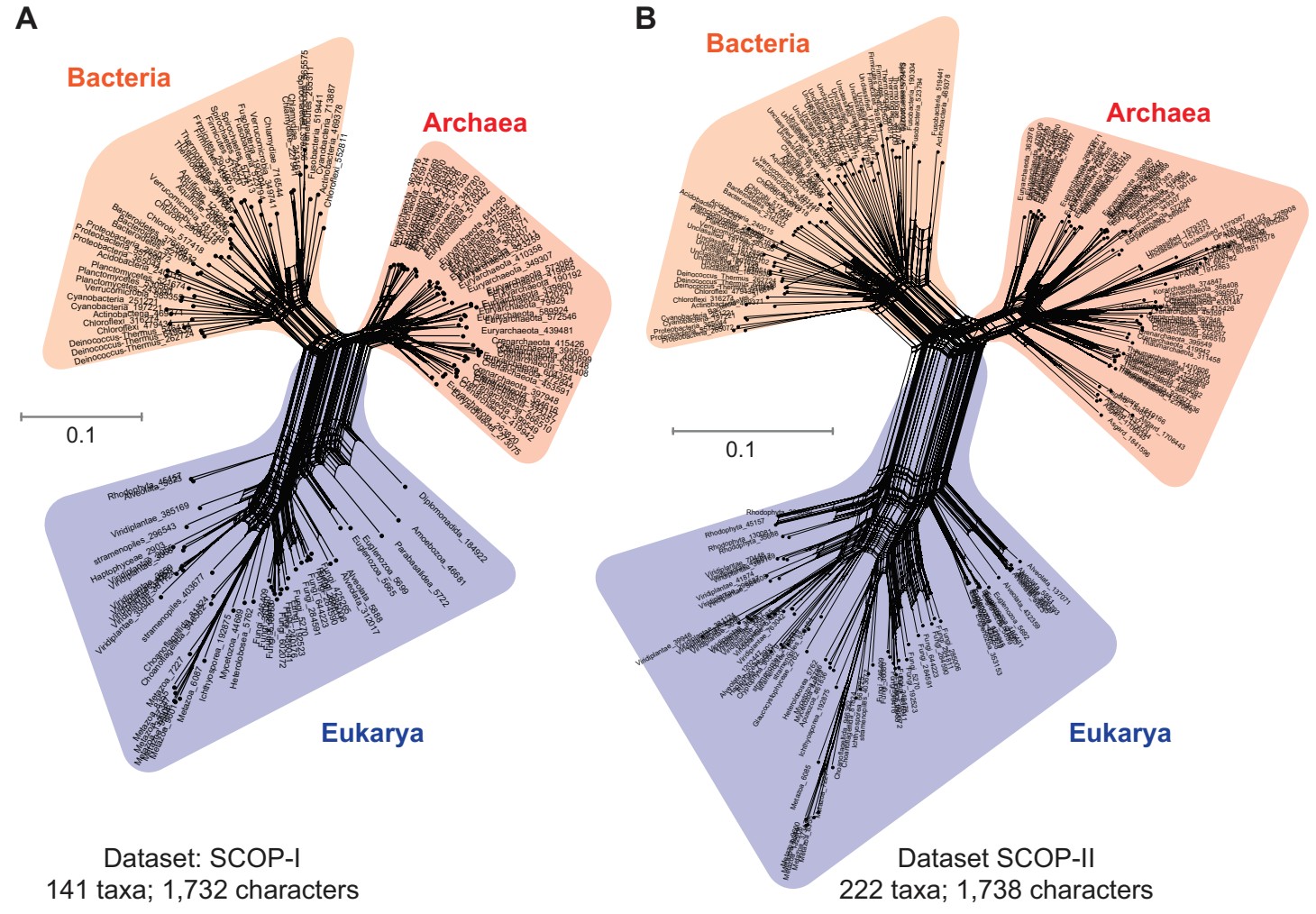

Dataset: SCOP-I
141 taxa; 1,732 characters

Dataset SCOP-II
222 taxa; 1,738 characters

**Figure 2 Data-display networks (DDN) depicting character conflicts among complex molecular characters.** Complex characters here are genomic loci that correspond to protein-domains as opposed to elementary characters (individual nucleotides or amino acids). The presence–absence patterns of homologous protein-domains identified by the structural classification of proteins (SCOP) scheme were coded with non-arbitrary state labels to assemble a data matrix. Each network is constructed from a neighbor-net analysis based on the Hamming distance identical to *p*-distance in (Fig. 1) and displayed as an equal angle split network. (A) DDN of 1,732 characters sampled from 141 species, each from distinct genera (SCOP-I dataset). (B) DDN based on an updated SCOP-I data matrix to include recently described novel species of Archaea and Bacteria, totaling to 222 species and a modest increase to 1,738 characters (SCOP-II dataset). Details of the DDNs are as in Fig. 1.

Table 2). Genomic loci that can be aligned with high confidence using MSA algorithms are typically more conserved than those loci for which alignment uncertainty is high. Such ambiguously aligned regions of sequences are routinely trimmed off before phylogenetic analyses (*Criscuolo & Gribaldo, 2010*). Typically, the conserved well-aligned regions correspond to protein-domains with highly ordered 3D structures with specific 3D folds (Fig. 3B). Accordingly, the MSA in core-genes-I dataset corresponds to 35 distinct domains found in 29 genes (Table 2), while 50 distinct domains are found in 48 genes sampled in the core-genes-II dataset. In the core-genes-I dataset, the number of unique domains (or loci) sampled per species varies between 25 and 35, since not all

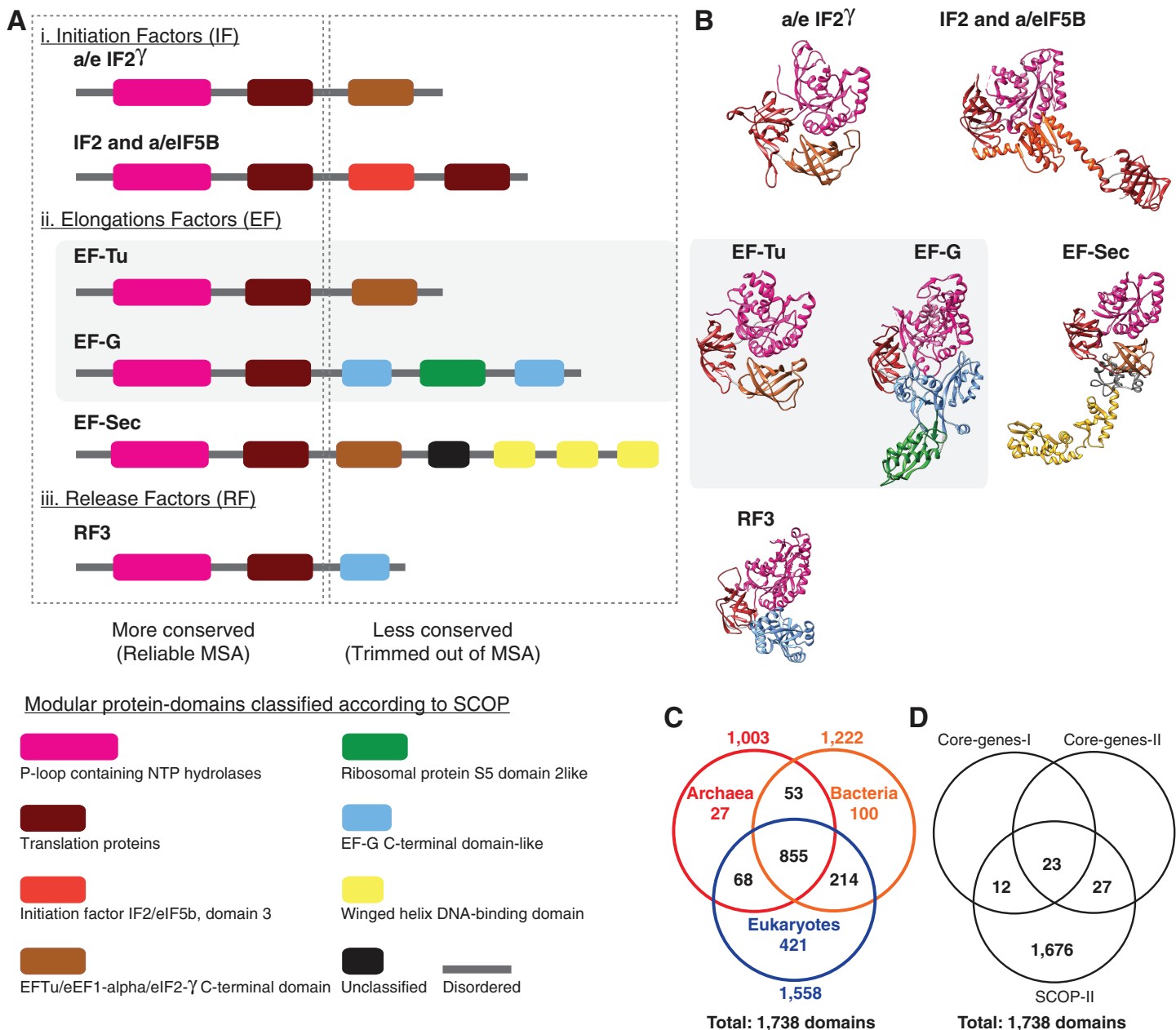

**Figure 3 Alignment uncertainty in closely related proteins due to domain recombination.** Multi-domain architecture (MDA), the N- to C-terminal sequence of the translational GTPase superfamily based on recombination of eight modular domains is shown as (A) linear sequences and (B) 3D structures. A total of 57 distinct families with varying MDAs are known, of which six canonical families are shown as a schematic in (A) and the corresponding 3D folds in (B). Amino acid sequences of only two of the eight conserved domains can be aligned with confidence for use in MSA-based phylogenomics. The length of the alignment varies from ~200–300 amino acids depending on the sequence diversity sampled (*Atkinson, 2015*; *Gouy, Baurain & Philippe, 2015*). The EF-Tu—EF-G paralogous pair employed as pseudo-outgroups for the classical rooting of the rRNA tree is highlighted. (C) Phyletic distribution of 1,738 out the 2,000 distinct SCOP-domains sampled from 222 species used for phylogenetic analyses in the present study. About 70% of the domains are widely distributed across the sampled taxonomic diversity. (D) Comparison of the number of genomic loci represented in the different data matrices used in phylogenomic studies.           

**Table 2 Redundant representation of genomic loci (protein-domains) in concatenated core-genes datasets.**

| Dataset | Number of taxa | Number of unique genes | Number of unique domains | Redundant domains | | Number of times redundant in each taxon | Number of taxa in which redundant |
|---|---|---|---|---|---|---|---|
| Core-genes-I | 44 | 29 | 35 | SCOP Unique ID | Description | | |
| | | | | 52540 | P-loop containing NTP hydrolases | 9 | 33 |
| | | | | | | 8 | 4 |
| | | | | | | 7 | 3 |
| | | | | | | 6 | 2 |
| | | | | | | 5 | 2 |
| | | | | 50447 | Translation proteins | 4 | 8 |
| | | | | | | 3 | 26 |
| | | | | | | 2 | 9 |
| | | | | 54211 | Ribosomal protein S5 domain 2-like | 3 | 42 |
| | | | | | | 2 | 2 |
| | | | | 50249 | Nucleic acid-binding proteins | 3 | 38 |
| | | | | | | 2 | 5 |
| | | | | 53067 | Actin-like ATPase domain | 2 | 16 |
| | | | | 54980 | EF-G C-terminal domain-like | 2 | 42 |
| | | | | 64484 | Beta and beta-prime subunits of DNA dependent RNA-polymerase | 2 | 41 |
| | | | | 47364 | Domain of the SRP/SRP receptor G-proteins | 2 | 2 |
| Core-genes-II | 96 | 48 | 50 | 50249 | Nucleic acid-binding proteins | 5 | 3 |
| | | | | | | 4 | 81 |
| | | | | | | 3 | 11 |
| | | | | | | 2 | 1 |
| | | | | 50104 | Translation proteins SH3-like domain | 3 | 78 |
| | | | | | | 2 | 18 |
| | | | | 50447 | Translation proteins | 3 | 15 |
| | | | | | | 2 | 71 |
| | | | | 64484 | Beta and beta-prime subunits of DNA dependent RNA-polymerase | 3 | 88 |
| | | | | | | 2 | 5 |
| | | | | 52540 | P-loop containing NTP hydrolases | 2 | 83 |
| | | | | 53067 | Actin-like ATPase domain | 2 | 40 |
| | | | | 53137 | Translational machinery components | 2 | 90 |
| | | | | 54211 | Ribosomal protein S5 domain 2-like | 2 | 93 |
| | | | | 56053 | Ribosomal protein L6 | 2 | 88 |
| SCOP-I | 141 | – | 1,732 | – | – | – | – |
| SCOP-II | 222 | – | 1,738 | – | – | – | – |

Note:
The P-loop NTP hydrolase domain is one of the most prevalent domains. Genomic loci encoding P-loop hydrolase domain are represented 5–9 times in each species in the single-copy genes employed in the core-genes datasets. Redundant loci in the core-genes datasets vary depending on the genes and species sampled for phylogenomic analyses. In contrast, SCOP-domain datasets are composed of unique loci.

loci are found in all species. While some loci are absent in some species, some loci are redundant. For instance, the P-loop nucleoside triphosphate (NTP) hydrolase domain, one of the most prevalent protein-domains, is represented up to nine times in many species (Table 2). Many central cellular functions are driven by the conformational changes in proteins induced by the hydrolysis of NTP catalyzed by the P-loop domain (*Chothia et al., 2003*).

Out of a total of 35 distinct domains in the core-genes-I dataset, seven are redundant, with two or more copies represented per species. Similarly, nine of the 50 domains have a redundant representation in the core-genes-II dataset (Table 2). The observed redundancy of the genomic loci in the MSA of core-genes is inconsistent with the common (and typically untested) assumption of using single-copy genes as a proxy for orthologous loci sampled for phylogenetic analysis. In contrast, the protein-domain datasets are composed of unique loci (Fig. 3C; Table 2). Further, the loci represented in the core-genes datasets make up only about 3% of the loci analyzed in SCOP-domain datasets in terms of the number of unique genomic loci sampled (Fig. 3D; Table 2).

Furthermore, regions of sequences that are filtered out, usually show higher variability in length, are less ordered and are known to accumulate insertion and deletion (indel) mutations at a higher frequency than in the regions that correspond to folded domains (*Light et al., 2013*; *Wang, Kurland & Caetano-Anollés, 2011*). These variable, structurally disordered regions, which flank the structurally ordered domains, link different domains in multi-domain proteins (Fig. 3A). Multi-domain architecture (MDA), the N- to C-terminal sequence of domain arrangement, is distinct for a protein family, and differs in closely related protein families with similar functions (Fig. 3A). The variation in MDA also relates to alignment uncertainties. Taken together, there is a major loss of information when core-genes datasets are employed for phylogenomic analyses compared to the protein-domain datasets (Figs. 3C and 3D) Information loss is due to:

  i) The small number of loci selected to start with in the core-genes datasets; at most 50 (Figs. 1 and 3D) compared to 1,738 in the SCOP-domain dataset (Figs. 2 and 3C), and,

 ii) The trimming of regions within MSAs due to alignment uncertainties (Fig. 3A).

Despite the relatively small number of characters that can be scored (~2,000), the protein-domains datasets (Fig. 2) are more informative to resolve the major taxa than the core-genes datasets (Fig. 1), for which a large number of characters are scored (~10,000).

## Data quality affects model complexity required to explain phylogenetic datasets

Resolving the monophyly or paraphyly of the Archaea is relevant to determining whether the three-domains tree (Fig. 4A) or the Eocyte tree (Fig. 4B), respectively,

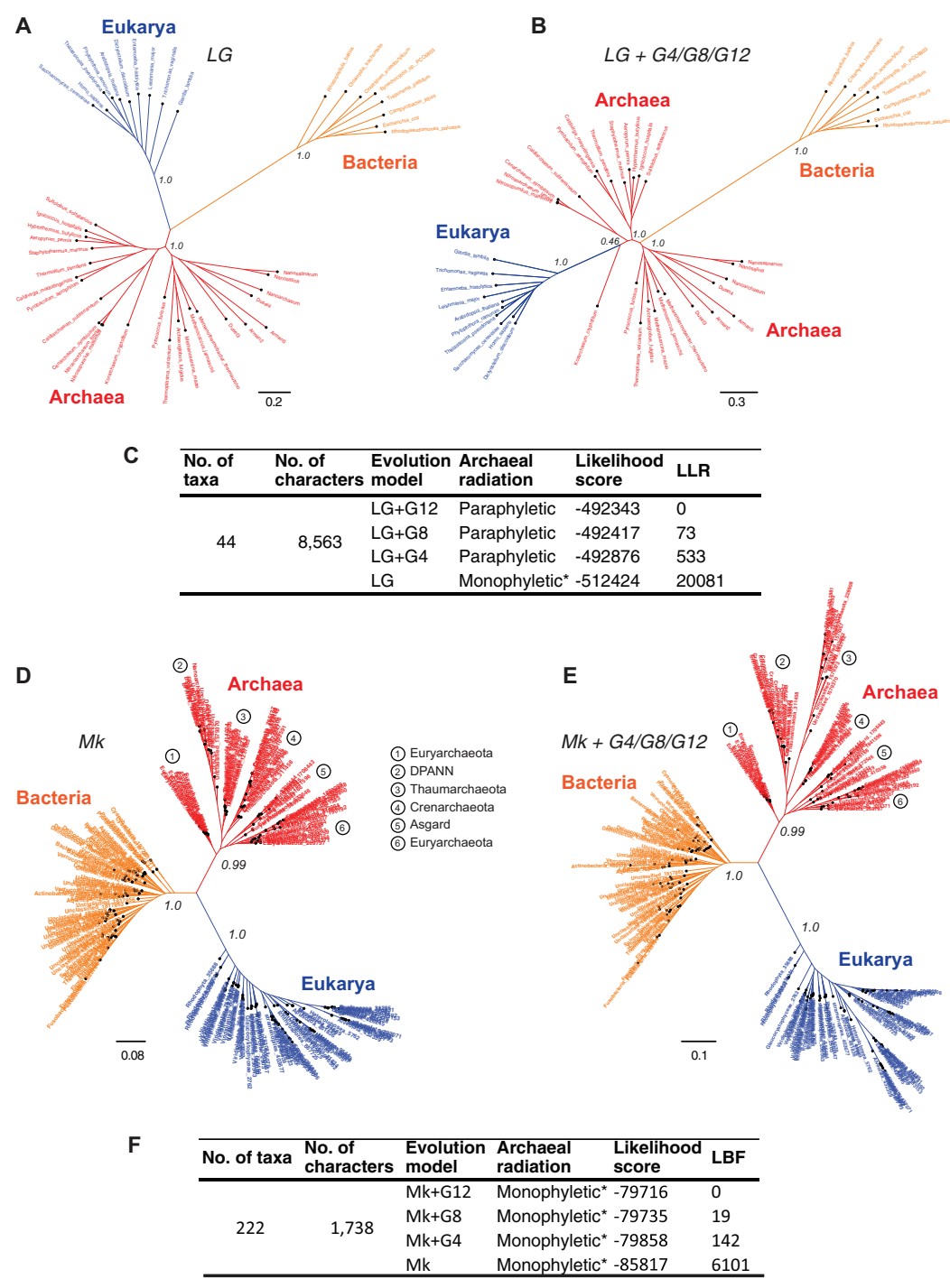

**A** Eukarya  *LG*  Bacteria  Archaea  0.2

**B** *LG + G4/G8/G12*  Archaea  Bacteria  Eukarya  Archaea  0.3

**C**

| No. of taxa | No. of characters | Evolution model | Archaeal radiation | Likelihood score | LLR |
|---|---|---|---|---|---|
| 44 | 8,563 | LG+G12 | Paraphyletic | -492343 | 0 |
| | | LG+G8 | Paraphyletic | -492417 | 73 |
| | | LG+G4 | Paraphyletic | -492876 | 533 |
| | | LG | Monophyletic* | -512424 | 20081 |

**D** *Mk*  Archaea  Bacteria  Eukarya  0.08

① Euryarchaeota
② DPANN
③ Thaumarchaeota
④ Crenarchaeota
⑤ Asgard
⑥ Euryarchaeota

**E** *Mk + G4/G8/G12*  Archaea  Bacteria  Eukarya  0.1

**F**

| No. of taxa | No. of characters | Evolution model | Archaeal radiation | Likelihood score | LBF |
|---|---|---|---|---|---|
| 222 | 1,738 | Mk+G12 | Monophyletic* | -79716 | 0 |
| | | Mk+G8 | Monophyletic* | -79735 | 19 |
| | | Mk+G4 | Monophyletic* | -79858 | 142 |
| | | Mk | Monophyletic* | -85817 | 6101 |

**Figure 4 Comparison of the sensitivity of the tree topology to character-specific rate heterogeneity (CSRH).** (A–C) Concatenated gene trees derived from amino acid characters, and (D–F) genome trees derived from protein-domain characters. (A, B) Unrooted trees estimated using the core-genes-I dataset for which (A) rate homogeneous-LG model, or (B) a CSRH-LG substitution model was implemented. Branch support values are approximate likelihood-ratio test (aLRT) scores (C) Model-fit to data is ranked according the log likelihood ratio (LLR) scores for the tree topology. LLR scores are computed as the difference from the best-fitting model (LG+G12) of the likelihood scores estimated in PhyML. Thus,

larger LLR values indicate lesser support for that model/tree, relative to the most-likely model/tree. Substitution rate heterogeneity is approximated with four, eight, or 12 rate categories in the complex models, but with a single rate category in the simpler model. (D, E) Genome trees derived from the SCOP-II datasets using (D) a rate homogeneous- or, (E) CSRH model of evolution of genomic protein-domain cohorts. Scale bars represent the estimated number of character-state changes. Branch support values are posterior probability (PP) scores estimated in MrBayes. (F) Model fit to data is ranked according log Bayes factor (LBF) scores, which like LLR scores are the log odds of the hypotheses. LBF scores are computed as the difference in likelihood scores estimated in MrBayes. Note: * Monophyly of Archaea is conditional on the placement of the root of the tree (see Fig. 5).

is a better-supported hypothesis. The Archaea are consistent with a monophyletic group in trees derived from a relatively simpler, rate-homogeneous LG model applied to the core-genes-I dataset (Fig. 4A). However, the Archaea are consistent with a paraphyletic group in trees derived from the more complex rate-heterogeneous versions of the LG model (Fig. 4B). In general, complex models tend to fit the data better. According to model selection tests for the core-genes-I dataset, the more complex versions of the LG model are better-fitting models than the simpler homogeneous-LG model (Fig. 4C; Table S2). Complex models account for various patterns of heterogeneity in amino acid substitutions. For instance, CSRH is accounted for by incorporating multiple rate-categories in the model. Substitution rate heterogeneity across different characters was approximated using a discrete Gamma distribution with four, eight, or 12 rate categories (LG+G4, LG+G8, or LG+G12, respectively). Model fit to data improves with the increase in complexity of the substitution model (Fig. 4C). Model complexity increases with any increase in the number of rate categories and/or the associated numbers of parameters that need to be estimated. Accordingly models that incorporate invariant sites (LG+GX+I) or MSA-specific state frequencies (LG+GX+F) and several combinations there of are even more complex. Recovering the Eocyte tree typically requires implementing complex models of sequence evolution rather than their relatively simpler (but over-simplified) versions (*Williams & Embley, 2014*). However, implementing more complex models did not change the tree topology (Fig. 4B) despite improved model fit to data (Fig. 4C; Table S2).

In contrast, trees inferred from the protein-domain datasets are consistent with monophyly of the Archaea irrespective of the complexity of the underlying model, with respect to CSRH (Figs. 4D–4F). The Mk model is the best-known probabilistic model of discrete character evolution for complex characters coded as binary-state characters (*Lewis, 2001*; *Wright & Hillis, 2014*). Since the Mk model assumes a stochastic process of evolution, it is able to estimate multiple state changes along the same branch. Both a simpler rate-homogeneous version of the Mk model (Fig. 4D), as well as more complex rate-heterogeneous versions with four, eight, or 12 rate categories (Mk+G4, Mk+G8, or Mk+G12, respectively) recovered trees that are consistent with the monophyly of the Archaea (Fig. 4E). The most complex model, Mk+G12 is the best-fitting model as seen from the LBF scores. A difference in LBF scores in the range of 3–5 is typically considered strong evidence in favor of the better model and topological hypothesis; while LBF difference of above five is considered very strong empirical evidence

(*Bergsten, Nilsson & Ronquist, 2013*; *Kass & Raftery, 1995*). The tree derived from the Mk+G4 model is shown in Fig. 4E. While the tree derived from Mk+G8 model is identical to the Mk+G4 tree, the Mk+G12 tree is almost identical with minor differences within the bacterial groups (see Fig. S1). This is likely to be due to the relatively more diverse set of species sampled from unclassified groups, and hence a low-density coverage of taxonomic groups within the Bacteria. However, species sampling from Archaea and Eukarya is relatively denser amongst taxonomic groups.

In all cases, bipartitions for Archaea show strong support with posterior probability (PP) of 0.99 while that of Bacteria and Eukarya is supported with a PP of 1.0—in spite of substantially different fits of the model to the data. A notable exception to the sequence-based classification is that the traditional phylum Euryarchaeota is not supported in this tree. Paraphyly of Euryarchaeota has also been observed with core-genes and single-gene datasets that were corrected for rate heterogeneity (*Foster, Cox & Embley, 2009*; *Gouy, Baurain & Philippe, 2015*). Nonetheless, the tree topology suggests that the Archaea is a distinct group. Even though the unrooted trees in Figs. 4A, 4D and 4E suggest monophyly of Archaea, verification of the unique evolutionary history of Archaea, or for that matter any other taxonomic group in the tree, depends on the placement of the root of the tree. Resolving the root of the global ToL is a difficult problem, both conceptually as well as technically, which is unlike other phylogenetic problems (see next section).

## Siblings and cousins are indistinguishable when reversible models are employed

An unrooted tree derived from standard reversible evolution-models is oblivious to the root, and thus has no evolutionary direction (Figs. 5A and 5B). Therefore an unrooted (undirected) tree is uninformative about: (1) ancestor-descendant polarity of taxa; (2) branching order; (3) evolutionary groups (or clades); and (4) ancestral and derived states. Given that a primary objective of phylogenetic analyses is to identify clades and the relationships between these clades, it is not possible to interpret an unrooted tree meaningfully without rooting the tree (see Fig. 5). Identification of clades as well as inferences of relationships between clades depends on the placement of the root or on prior assumptions about the root. In general, it is not possible to make evolutionary inferences from any unrooted (undirected) topological hypothesis.

For instance, although a DDN is useful to diagnose character conflicts in phylogenetic datasets and to postulate evolutionary hypotheses, a DDN by itself cannot be interpreted as an evolutionary network, because the edges do not necessarily represent evolutionary phenomena and the nodes do not represent ancestors (*Huson & Bryant, 2006*; *Morrison, 2009*). Therefore, evolutionary relationships cannot be inferred from a DDN. Likewise, evolutionary relationships cannot be inferred from unrooted trees, even though nodes in an unrooted tree do represent ancestors and an evolution-model defines the branches (Figs. 5A and 5B). An unrooted tree is consistent with more than one rooted tree (Fig. 5).

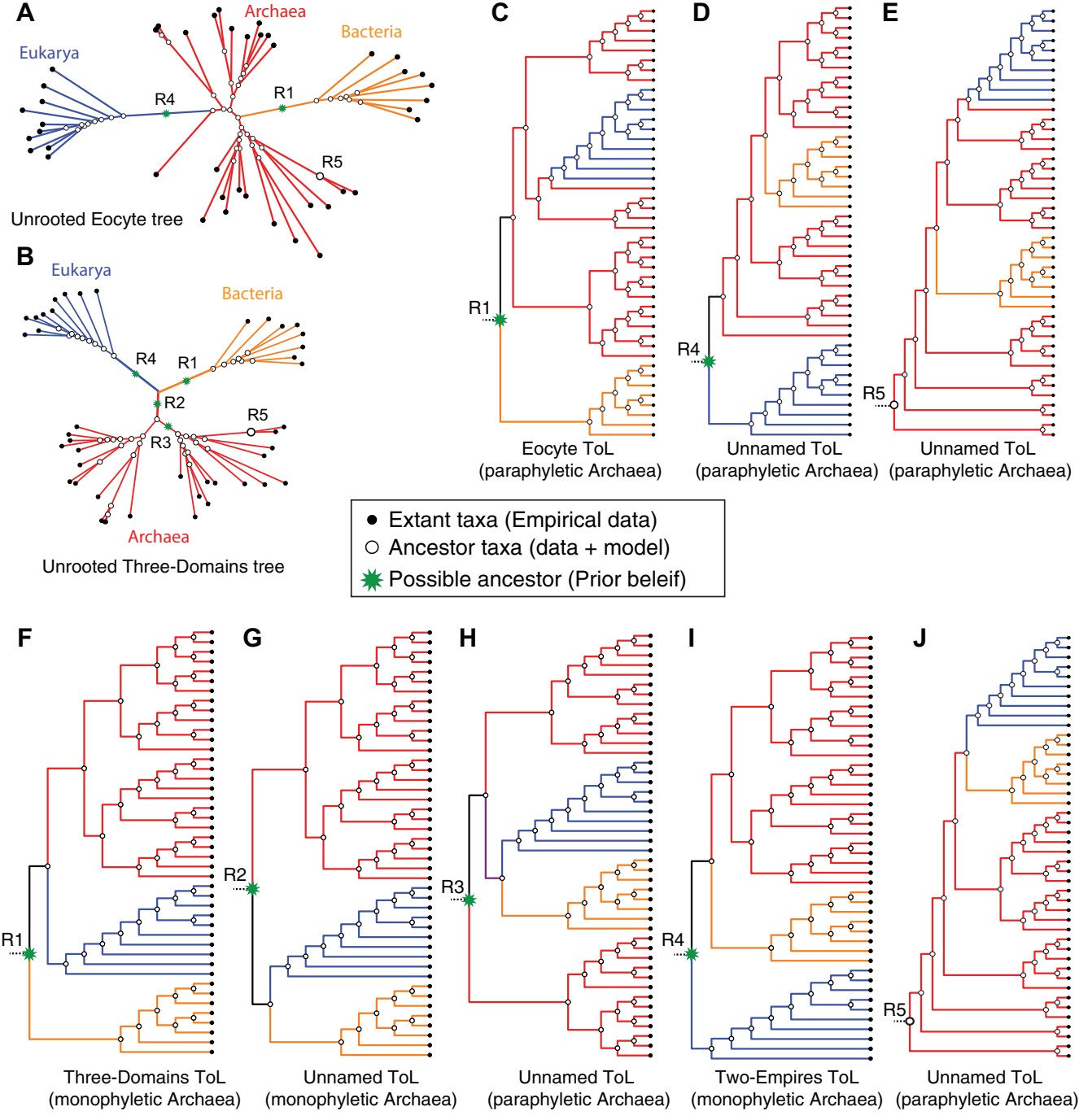

**Figure 5 Impact of alternative *ad hoc, a posteriori* rootings on the phylogenetic classification of archaeal biodiversity.** (A, B) Unrooted trees derived from standard evolution-models are oblivious to the root and are not fully resolved into bipartitions (i.e., some braches are polytomous rather than dichotomous), and thus preclude identification of clades and sister group relationships. With multiple, independent sets of bipartitions, the Archaea are unresolved in (A), but are resolved into a distinct set of bipartitions in (B). It is common practice to add a user-specified root node (green*) *a posteriori* to unrooted trees, by hand, based on prior knowledge (or belief) of the investigator. Such an *a posteriori* rooting is necessary to determine the recency of common ancestry as well as the temporal order of key evolutionary transitions that define evolutionary groups. Five possible (of many) rootings R1–R5 are shown (see text for description). (C–J) The different possible evolutionary relationships of the Archaea to other taxa, depending on the position of the root, are shown. Both the Eocyte ToL (A) and the three-domains ToL (F) depend on the notion that the root should be placed at position R1 in the unrooted tree. (I) Two-empires ToL based on the root placed at position R4. (D, E, G, H, and J) arbitrarily rooted ToL.

The identity of the root corresponds, in principle, to any one of the possible hypothetical ancestors as follows:

i) Any one of the inferred-ancestors at the resolved bipartitions (open circles in Figs. 5A and 5B), or

ii) Any one of the yet-to-be-inferred-ancestors that lies along the stem-branches of the unresolved polytomy (green stars Figs. 5A and 5B) or along the internal-braches.

In the latter case, rooting the tree *a posteriori* on any of the branches amounts to inserting an additional bipartition and an ancestor that is neither inferred from the source data nor deduced from the underlying character evolution-model. Hence rooting, and interpreting the ToL depends on:

i) Prior knowledge—for example, fossils or known sister group (outgroup) taxa, or

ii) Prior beliefs/expectations of the investigators—for example, simple is primitive (*Nasir & Caetano-Anollés, 2015*; *Whittaker, 1969*), Bacteria are primitive (*Sagan, 1967*; *Stanier & Van Niel, 1962*), Archaea are primitive (*Woese & Fox, 1977*), etc.

Both of these options are independent of the data used to infer the unrooted ToL. Accordingly, both the three-domains hypothesis and the Eocyte hypothesis depend on the notion that the root should be placed on the stem branch leading to the Bacteria (root R1 in Fig. 5) in the unrooted tree. Other possible rootings and the resulting rooted-tree topologies are shown in Figs. 5C–5J. In the unresolved tree (Figs. 4B and 5B) Archaea would be paraphyletic irrespective of the placement of the root. In all other cases (Figs. 4A, 5B and 5C), if the root lies on any of the internal branches (e.g., R3 in Figs. 5A and 5B), or corresponds to one of the internal nodes within the archaeal radiation (e.g., R5 in Figs. 5A and 5B), the Archaea would not constitute a unique clade (Fig. 5). However, if the root lies on one of the stem branches (R1/R2/R4 in Fig. 5B), monophyly of the Archaea would be unambiguous (Figs. 5F, 5G and 5I). Determining the evolutionary relationship of the Archaea to other taxa, though, requires identifying the root.

The common practice of *a posteriori* rooting, that is, converting an unrooted (undirected) ToL into a rooted (directed) ToL, *by hand*, implies prior knowledge of the polarity of character transitions from ancestral-to-derived states. In other words, prior knowledge of the ancestral (root) states of characters is necessary to root a tree, which is commonly inferred from outgroup taxa. In the absence of prior knowledge of the root, directional evolution-models are useful for identifying the root (*Huelsenbeck, Bollback & Levine, 2002*; *Klopfstein, Vilhelmsen & Ronquist, 2015*; *Yang & Roberts, 1995*). Unlike reversible models, directional models are able to identify the polarity of state transitions, and thus the root of a tree. Moreover, directional models are useful to evaluate the empirical support for prior beliefs about the universal common ancestor (UCA) at the root of the ToL (*Harish & Kurland, 2017a*). Directional evolution refers to two distinct, but related aspects of the evolutionary process, non-reversibility and non-stationarity (*Harish & Kurland, 2017b*; *Klopfstein, Vilhelmsen & Ronquist, 2015*). Non-reversibility refers to the asymmetric propensity of character transitions, that is, propensity for change

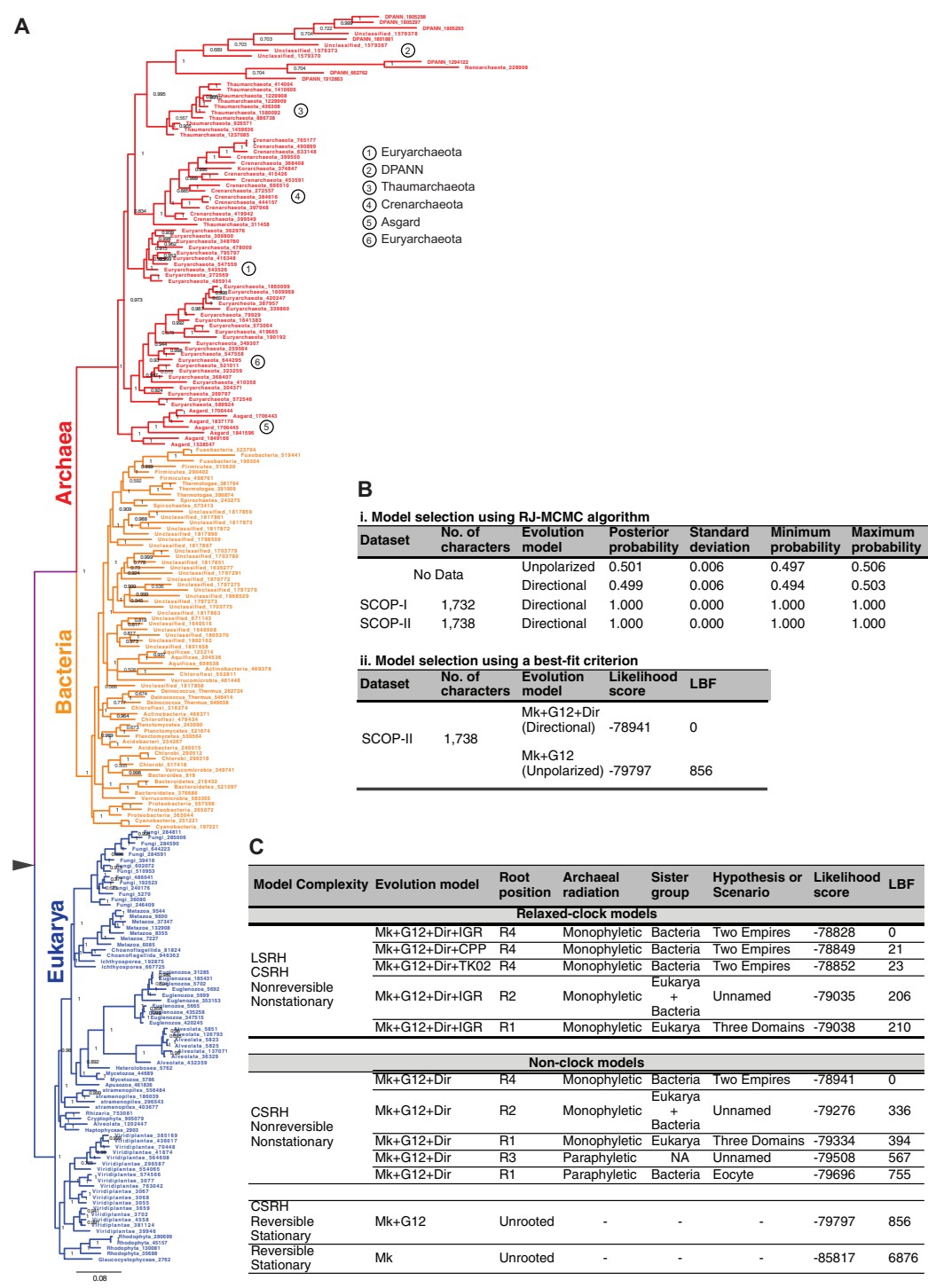

**Figure 6 Global tree of life depicting the evolutionary relationships of the major taxa of life.** (A) Phylogeny of the major taxa Archaea, Bacteria, and Eukarya inferred from patterns of inheritance of functional genomic signatures. Monophyly of each major taxon and placement of Archaea sister to Bacteria supports a dichotomous classification of the diversity of life such that Archaea and Bacteria together constitute a clade of akaryotes (or Akarya). Eukarya and Akarya are sister-clades that diverge from the universal common ancestor (UCA) at the root of the tree of life. Each clade is supported by the highest posterior probability of 1.0. The phylogeny supports a scenario of independent origins and

descent of eukaryote and akaryote species. (B) Model selection tests identify, overwhelmingly, directional evolution-models to be better-fitting models to describe the evolution of genomic signatures. (C) The estimated phylogeny, especially the placement of the root is robust to both CSRH and LSRH. Alternative hypotheses, and accordingly alternative classifications or scenarios for the origins of the major clades of life, are much less probable and not supported.

from one state to another along the tree is different from a change in the reverse direction. Non-stationarity refers to change in frequencies of characters (or states) in evolutionary time when conditions within a group differs from the conditions in its sister groups and thus at the root of the tree. Taken together, directional evolution refers to a non-random, and non-reversible shift in the marginal distribution of traits during evolutionary time (*Klopfstein, Vilhelmsen & Ronquist, 2015*).

The Bayesian model selection test (implemented in MrBayes) to detect directional trends chooses the directional model (Fig. 6B), overwhelmingly, over the reversible model for the SCOP-II dataset (Fig. 2B). Further, the best-supported rooting corresponds to root R4 (Figs. 5G, 6A and 6C). Monophyly of the Archaea is maximally supported (PP of 1.0). Furthermore, this rooting places Archaea sister to Bacteria with the highest support (PP 1.0). Accordingly, a higher order taxon, akaryotes, proposed earlier (*Forterre, 1992*) forms a clade with robust support (PP 1.0). Thus akaryotes (or Akarya) and Eukarya are sister clades that diverge from the UCA at the root of the ToL. Identical results were obtained for the SCOP-I dataset (Fig. 2A) as reported previously (*Harish & Kurland, 2017a*). The placement of the root as well as the tree topology is robust against long-branch attraction (LBA) artifacts due to CSRH and LSRH (Fig. 6C).

The simplest model, the standard stationary and reversible evolution-model, is the worst fitting model far and away (Fig. 6C). As such, complex models that account for non-reversibility, non-stationarity, CSRH, and LSRH are the better-fitting models. In all cases, and regardless of model complexity, root R4 is the best-supported rooting (Fig. 5I). Thus the two-empires of life hypothesis (*Mayr, 1998*) is the best-supported ToL (Figs. 6A and 6C). Alternative rootings are much less likely, and therefore other ToL hypotheses are not supported (Fig. 6C). Accordingly, independent origin of eukaryotes as well as akaryotes is the best-supported scenario. The three-domains ToL (root R1; Fig. 5F) is $10^{171}$ times less likely, and the scenario proposed by the Eocyte ToL (root R1, Fig. 5C) is highly unlikely, as are other scenarios (root R2, R3; Figs. 5G–5H). The traditional belief that simple is primitive, as well as beliefs that Archaea are primitive or that Archaea and Bacteria evolved before eukaryotes, are not supported either.

## DISCUSSION

### Improving data quality can be more effective for resolving recalcitrant branches than increasing model complexity

A diversity of evolutionary signatures in molecular sequence data is utilized by different analytical approaches to recover phylogenetic signal. Recovery of phylogenetic signal in sequence-alignment data by the analysis of variation in single-residue substitutions is the predominant (standard) approach. Other less frequently used sources of phylogenetic
signal includes variation in triplet-codons, multiresidue indels in protein-coding and non-coding loci as well as variation in the gain, loss, and copy number of the loci per se (*Harish & Kurland, 2017b*; *Hillis, 1999*; *Rokas & Holland, 2000*). In the phylogenetic literature, the concept of data quality refers to the quality or the strength of the phylogenetic signal that can be recovered from the data. The strength of the phylogenetic signal is proportional to the confidence with which unique state-transitions can be determined for a given set of characters on a given tree. Ideally, historically unique character transitions that entail rare evolutionary innovations are desirable to identify patterns of uniquely shared innovations (synapomorphies) among lineages. Synapomorphies are the diagnostic features used for assessing lineage-specific inheritance of evolutionary innovations. Therefore identifying character transitions that are likely to be low probability events is crucial for the accuracy of phylogenetic analysis.

In their pioneering studies, Woese and colleagues identified unique features of the SSU rRNA—"oligonucleotide signatures"—to determine evolutionary relationships (*Pechman & Woese, 1972*; *Woese & Fox, 1977*). An underlying assumption was that the probability of occurrence of the same set of oligomer signatures by chance, in non-homologous sequences, is low in a large molecule like SSU rRNA (1,500–2,000 nucleotides). Further, the study found that only oligomers that were six nucleotides or longer were robust markers of homology of the SSU rRNA. Oligomers shorter than six nucleotides were statistically less likely to be efficient markers of homology (*Pechman & Woese, 1972*; *Woese et al., 1975*). Thus larger oligomer signatures are more robust characters than shorter oligomers or monomers to determine a phylogeny of the SSU rRNA gene (or gene family).

However, as sequencing of full-length rRNAs and statistical models of nucleotide substitution became common, complex oligomer-characters were replaced by elementary nucleotide-characters; and more recently by amino acid characters (Fig. 1). Identifying rare or historically unique substitutions in empirical datasets has proven to be difficult (*Parker et al., 2013*; *Rokas & Carroll, 2008*), consequently the uncertainty of resolving the deeper branches of the ToL using marker gene-sequences remains high. A primary reason is the relatively higher prevalence of phylogenetic noise (homoplasy) in primary sequence datasets (Fig. 1), due to the characteristic redundancy of nucleotide and amino acid substitutions and the resulting difficulty in distinguishing phylogenetic noise from signal (homology) (*Philippe et al., 2011*; *Rokas & Carroll, 2006*). Better-fitting (or best-fitting) models are expected to recover phylogenetic signal more efficiently and thus explain the data better, but tend to be more complex than worse fitting models (*Lartillot & Philippe, 2004*; *Williams & Embley, 2014*).

Unrooted topologies estimated from the core-genes-I dataset using CSRH-LG models are congruent (Fig. 4B), in spite of the significantly different model-fits to the data (Fig. 4C). Likewise, both unrooted- (Figs. 4D and 4E) and rooted (Fig. 6A) topologies estimated from the SCOP-II dataset using ~15 distinct models of increasing complexity, and significantly different model-fits to data (Figs. 4E and 6C), are largely congruent. These results show that increased model complexity, or improved model-fit to data, does not necessarily resolve conflicting signals in phylogenetic datasets. Although increasing model complexity can correct errors of estimation and improve the fit of the data to the

tree, it is not a solution to improve phylogenetic signal, especially when the historical signal is exceedingly limited or absent in the source data (Figs. 1, 3C and 3D; Table 2).

The idea of "oligonucleotide signatures" used for estimating a gene phylogeny has been extended, naturally, to infer a genome phylogeny (Graham et al., 2000). The signatures were defined in terms of protein-coding genes that were shared among the Archaea. However, as proteins are recombinant-mosaics of domains, domains are unique genomic signatures (Fig. 3). Protein-domains identified by SCOP correspond to complex "multidimensional signatures" defined by: (i) a unique 3D fold, (ii) a distinct sequence profile, and (iii) a characteristic function. Though domain recombination is frequent, substitution of one protein-domain for another has not been observed in homologous proteins (Fig. 3). Thus, protein-domains are "functional genomic signatures." For phylogenomic applications, protein-domains are "sequence signatures" that essentially correspond to single-copy orthologous loci when coded as binary-state (presence–absence) characters. These sequence signatures are consistent with unique, non-recombining genomic loci, and are identified using sophisticated statistical models—profile hidden Markov models (Eddy, 1998; Park et al., 1998)—that can be used routinely to annotate and curate genome sequences in automated pipelines (Fang et al., 2013; Gough et al., 2001).

For these reasons, protein-domains are highly effective phylogenetic markers for which character homology can be validated through more than one property: statistically significant (i) sequence similarity, (ii) 3D structure similarity, and (iii) function similarity. In addition, employing loci for protein-domains maximizes the genomic information that can be employed for phylogenetic analysis (Table 2; Figs. 3C and 3D). Even though many other genomic features are known to be useful markers (Rokas & Holland, 2000), protein-domains are the most conserved as well as most widely applicable genomic characters (Fig. 3C). Protein-domain characters are not without caveats (see Harish & Kurland, 2017b for a Discussion).

Character recoding is found to be effective in reducing the phylogenetic noise in primary sequence data (Susko & Roger, 2007). This is a form of data simplification wherein the number of amino acid alphabets is reduced to a smaller set of alphabets (usually from 20 to 6) that are frequently substituted for each other. Character recoding into reduced alphabets is useful to minimize phylogenetic artifacts such as LBA due to substitution saturation or compositional heterogeneity. However, character recoding does not reduce the noise in the core-genes-II dataset (Fig. 2D). Contrary to the expectation, there is an increase in the apparent noise, as seen from an increase in the extent of reticulation compared to the original (untreated) data (Fig. 2C). Common methods of estimating mutational saturation in sequences, particularly from multi-gene concatenations tend to underestimate the degree of saturation (Whelan et al., 2015). This seems to be the case especially for ribosomal proteins, which dominate the core-genes datasets (see Whelan et al., 2015 for a detailed characterization).

Therefore, datasets in which phylogenetic noise is inherently limited are more desirable, to minimize ambiguities. Like amino acids, protein-domains are also modular alphabets, albeit higher order, and more complex alphabets of proteins. Moreover, unlike the 20 standard amino acids, there are approximately 2,000 unique protein-domains identified

at present according to SCOP (*Murzin et al., 1995*). The number is expected to increase; theoretical estimates range between 4,000 and 10,000 distinct domain modules, depending on the classification scheme (*Govindarajan, Recabarren & Goldstein, 1999*). Coding homologous features as binary characters is the simplest possible representation of data for describing historically unique events. Accordingly, resolving character conflicts observed in the data (Fig. 2) would be less demanding, as such conceptually, and also computationally less expensive for large-scale empirical datasets.

## Sorting evolutionary signal from noise

Single-copy genes are employed as phylogenetic markers to minimize phylogenetic noise caused by reticulate evolution including hybridization, introgression, recombination, horizontal transfer (HT), duplication-loss (DL), or incomplete lineage sorting (ILS) of genomic loci. However, the noise observed in the DDNs based on MSA of core-genes (Fig. 1) cannot be directly related to any of the above genome-scale reticulations, since the characters are individual nucleotides or amino acids. Apart from stochastic character conflicts, the observed conflicts are better explained by convergent substitutions, given the redundancy of substitutions. Convergent substitutions caused either due to stringent selection or by chance are a well-recognized form of homoplasy in gene-sequence data (*Castoe, De Koning & Pollock, 2010*; *Philippe et al., 2011*; *Rokas & Carroll, 2008*), and based on recent genome-scale analyses it is now known to be rampant (*Foote et al., 2015*; *Liu et al., 2010*).

The observed noise in the DDNs based on protein-domain characters (Fig. 2), however, can be related directly to genome-scale reticulation processes and homoplasies. In general, homoplasy implies evolutionary convergence, parallelism, or character reversals caused by multiple processes. In contrast, homology implies only one process: inheritance of traits that evolved in the common ancestor and were passed to its descendants. Operationally, tree-based assessment of homology requires tracing the phylogenetic continuity of characters (and states), whereas homoplasy manifests as discontinuities along the tree. Since clades are diagnosed on the basis of shared innovations (synapomorphies) and defined by ancestry, accuracy of a phylogeny depends on an accurate assessment of homology (*Avise & Robinson, 2008*; *Hennig, 1965*; *Morrison, 2006*; *Padian, Lindberg & Polly, 1994*).

Identifying homoplasies caused by character reversals, that is, reversal to ancestral states requires identification of the ancestral state of the characters under study. However, implementing reversible models precludes the estimation of ancestral states, in the absence of sister groups (outgroups) or other external references. Thus, the crucial distinction between similarity due to homoplasy and homology as well as between shared ancestral homology (symplesiomorphy) and shared derived homology (synapomorphy) is not possible with unrooted trees derived from standard reversible models. Hence, unrooted trees (Fig. 4) are not evolutionary trees per se, as they are uninformative about the evolutionary polarity (*Morrison, 2006*; *Wiley & Lieberman, 2011*). Thus, identifying the root (or root-state) is crucial to (i) determine the polarity of state transitions, (ii) identify synapomorphies, and (iii) diagnose clades.

For complex characters such as protein-domains, character homology can be determined with high confidence using sophisticated statistical models (HMMs). Homology of a protein-domain implies that the de novo evolution of a genomic locus corresponding to that protein-domain is a unique historical event. Therefore, homoplasy due to convergences and parallelisms is highly improbable (*Mackin, Roy & Theobald, 2014*; *Pethica, Levitt & Gough, 2012*). Although a handful of cases of convergent evolution of 3D structures are known, these instances relate to relatively simple 3D folds coded for by relatively simple sequence repeats (*Mistry et al., 2013*). However, the vast majority of domains identified by SCOP correspond to polypeptides that are on average 200 residues long with unique sequence profiles (*Gough et al., 2001*; *Pethica, Levitt & Gough, 2012*). Thus, identifying homoplasy in the protein-domain datasets depends largely on estimating reversals, which will be cases of secondary gains and losses. For instance, reversals due to gain-loss-regain events caused by HT or DL-HT are homoplasies. Such secondary gains are more likely to relate to HT events than to convergent evolution, for reasons specified above. Instances of reversals are minimal, as seen from the strong directional trends detected in the data (Fig. 6B). Thus, employing complex molecular characters minimizes uncertainty in determining polarity of state transitions, identifying synapomorphies, and diagnosing clades.

Moreover, because clades are associated with the emergence and inheritance of evolutionary novelties, the discovery of clades is fundamental for describing and diagnosing sister group differences (*Sanderson, 2005*). A well-recognized deficiency of phylogenetic inference based on primary sequences is the abstraction of evolutionary "information": For instance, "information" relevant to diagnosing clades and support for clades is abstracted to branch lengths. Branch-length estimation is, ideally, a function of the source data and the underlying model. However, in the core-genes-I dataset the estimated branch lengths and the resulting tree is an expression of the model rather than of the data (Figs. 4A and 4B). Some pertinent questions then are: should diagnosis of clades and the features by which clades are identified be restricted to substitution mutations in a small set of loci and substitution models? Are substitution mutations in 40–50 loci more informative, or the evolution of unique genomic loci—functional genomic signatures—more informative?

Proponents of the total evidence approach recommend that all relevant information—molecular, biochemical, anatomical, morphological, fossils—should be used to reconstruct evolutionary history, yet genome sequences are the most widely applicable data at present (*Rokas & Holland, 2000*; *Wheeler, Assis & Rieppel, 2013*). Accordingly, phylogenetic classification is, in practice, a classification of genomes. There is no *a priori* theoretical reason that phylogenetic inference, especially of the global ToL, should be restricted to a small set of genomic loci corresponding to the core-genes, nor is there a reason for limiting phylogenetic models to interpreting patterns of substitution mutations alone. The ease of sequencing and the practical convenience of assembling large character matrices, by themselves, are no longer compelling reasons to adhere to the traditional marker gene-sequence analysis.

When phylogenetic inference is based on the protein-domain datasets, the gain and loss of distinct sets of "functional signatures" that define clades can be identified, which is unlike inferences based on core-genes datasets (see Supplementary Information in *Harish, Tunlid & Kurland, 2013*). Annotations for reference genomes of homologous protein-domains identified by SCOP and other protein classification schemes, as well as tools for identifying corresponding sequence signatures, are readily available in public databases. An added advantage is that the biochemical function and molecular phenotype of the domains are readily accessible as well, through additional resources including the protein data bank and InterPro (*Finn et al., 2016*).

## The critical distinction between gene-centered and genome-centered phylogenetic models

As mentioned in the previous section, assessment of homology is fundamental for inferring character evolution as well as evolutionary relationships between the operational taxonomic units (OTUs). Because OTUs are defined arbitrarily, the distinction between gene-OTUs and genome-OTUs cannot be emphasized enough. This distinction is crucial for the assessment of molecular homology, since homology is a hierarchical concept (*Dickinson, 1995*; *Morrison, Morgan & Kelchner, 2015*). Homology at one level in the hierarchy need not necessarily imply homology at another level of biological organization. Accordingly, homology at different levels is detected using different criteria.

For instance, homology of gene-OTUs in any given gene cluster is inferred from statistically significant overall-similarity of the genes without considering the homology of individual nucleotide or amino acid characters. Clusters of gene-OTUs are identified and classified into families based on measures of overall-similarity, which is estimated either as pairwise sequence similarity (e.g., BLAST) or similarity to sequence profiles (e.g., PSI-BLAST and HMM) (*Pearson & Sierk, 2005*).

Measures of overall-similarity do not distinguish between homologous similarity and similarity by chance (homoplasy) of individual characters. In contrast, phylogenetic methods have the distinct advantage of distinguishing the evolutionary signal of homology from the noise due to homoplasy (*Avise & Robinson, 2008*; *Morrison, 2006*). Even though determining the evolutionary polarity of character transitions is key to distinguish signal from noise, many commonly used tools of inference such as MSAs and unrooted trees are oblivious to the polarity of evolutionary transitions, and hence to the evolutionary path. This can often result in erroneous estimates and spurious placements of OTUs (*Eisen, 1998*; *Kurland, Canback & Berg, 2003*). That is to say that the nearest neighbor in an unrooted tree (or in an overall-similarity network) need not necessarily be the closest relative, as shown in Fig. 5. Identifying the root is critical even when the OTUs are individual genes to avoid misleading conclusions. This is decidedly relevant to phylogenomic analyses designed to identify clades and to determine trends in macroevolution (*Harish & Kurland, 2017b*, *2017c*).

It is becoming increasingly clear that conventional phylogenomic approaches frequently fail to resolve the deeper nodes of the ToL reliably (*Philippe et al., 2011*; *Shen, Hittinger & Rokas, 2017*; *Whelan et al., 2015*). Existing methods that rely on

recovering phylogenetic signal from MSAs neither identify, nor describe sister-group differences satisfactorily in spite of employing several hundreds of MSAs. A well-studied problem is the identification of the root of the Metazoa (animals), which is a relatively shallow node in the global ToL compared to the root of the Archaea. Efforts to resolve the metazoan-root have employed concatenated MSAs of up to 1,000 genes, and yet it remains ambiguous (*Philippe et al., 2011*; *Shen, Hittinger & Rokas, 2017*; *Whelan et al., 2015*). In comparison to the metazoan-root, the number of genes that can be aligned to the resolve the root of the global ToL is extremely limited, to approximately 50 (*Zaremba-Niedzwiedzka et al., 2017*).

One approach to overcome this limitation involves estimating trees of individual gene-families *en masse*. Topologies of individual gene-OTUs are summarized in order to estimate the support for the monophyly (unique origin) of major taxa: Archaea, Bacteria, and Eukarya (*Nelson-Sathi et al., 2015*; *Rochette, Brochier-Armanet & Gouy, 2014*; *Thiergart et al., 2012*). However, all of the trees are derived from reversible and stationary models, which yield unrooted trees. Therefore *ad hoc* sister groups (outgroups) are specified to determine the root and polarity of evolution. And, the choice of sister group(s) is itself based on measures of overall-similarity. In other words the identification of the origin (root) of any given gene family is *ad hoc* (Figs. 5A and 5B). Misidentification of sister groups along with spurious placements of gene-OTUs can potentially confound the interpretation of such unrooted trees (*Graham, Olmstead & Barrett, 2002*).

For instance, the classical rooting of the (rRNA) ToL based on the EF-Tu—EF-G paralogous pair (*Baldauf, Palmer & Doolittle, 1996*; *Iwabe et al., 1989*) is known to be error prone and highly ambiguous, due to systematic errors including LBA, compositional bias, and model misspecification (*Brinkmann & Philippe, 1999*; *Gouy, Baurain & Philippe, 2015*). Remarkably, sequences corresponding to only one of the two conserved domains common to EF-Tu and EF-G, the P-loop-containing NTP hydrolase domain (Fig. 3A) can be aligned with confidence. This single-domain MSA is ~200 residues in length. Implementing better-fitting substitution models results in two alternative rootings (*Gouy, Baurain & Philippe, 2015*). These are root R1 (on the branch leading to Bacteria) and a root within the Archaea that is similar to root R5 (Fig. 5B). These alternative rootings relate to distinct, irreconcilable scenarios. Further, the EF-Tu—EF-G paralogous pair is only two of 57 known paralogs of the translational GTPase protein superfamily (*Atkinson, 2015*). Thus the assumption that EF-Tu—EF-G duplication is a unique event, which is essential for the paralogous outgroup-rooting method, is untenable (Fig. 3). Furthermore, the root inferred for one gene (or domain) family may not be applicable to another family due to the prevalent discordance between individual gene trees, and between gene trees and species trees. Therefore, the Dayhoff duplicate-gene-rooting method (*Schwartz & Dayhoff, 1978*; *Woese, Kandler & Wheelis, 1990*) is not suitable to root genome trees or species trees.

These findings underscore the importance of acknowledging the hierarchical difference between genome-OTUs and gene-OTUs as well as relevant character evolution-models used to determine evolutionary relationships (*Boussau & Daubin, 2010*; *Coenye et al., 2005*; *Harish & Kurland, 2017b*). However, current phylogenomic approaches involve

analysis of a concatenated-MSA of highly conserved (core) genes on the one hand and independent analysis of less conserved (accessory) genes on the other. That is, contrasting approaches are applied to different regions of genomes that are conserved to different extents. Such contrasting treatments point to a rather obvious predicament. That is, if it is not possible to recover reliable phylogenetic signal from concatenated-MSAs, of many conserved *marker genes* (Fig. 1), how reliable is the signal from individual MSAs of relatively less conserved gene families? Not quite reliable, evidently, since existing MSA-analysis methods are unable to adequately distinguish phylogenetic signal and noise for the OTUs employed.

In other words, existing phylogenomic methods that employ reversible/stationary sequence evolution-models are suboptimal for determining the temporal order of key evolutionary transitions in the ToL. Therefore, inferences of the origin of individual gene families as well as the estimated evolutionary path are likely to be error prone. For instance, it is not possible to determine if anomalous placement of a gene-OTU is due to a lack of phylogenetic signal or HT without adequately distinguishing signal from noise (*Rochette, Brochier-Armanet & Gouy, 2014*). Accordingly, the origin of individual gene families may be untraceable from the analysis of single-gene MSAs using existing methods. This calls for an explicit distinction between evolutionary inferences drawn from qualitatively different gene-scale (Fig. 1) and genome-scale (Fig. 2) evolutionary signatures as well as qualitatively different (e.g., directional vs. reversible) evolution-models (see next section).

## Untangling data bias, model bias, and interpretation bias (prior beliefs)

A single class of genes, those encoding ribosomal proteins dominates core-genes datasets; for example ribosomal proteins make up 66% (32/48) of the core-genes-II dataset (*Zaremba-Niedzwiedzka et al., 2017*). Further, core-genes datasets predominantly relate to one functional class (Translation) of the ~25 functional classes assigned to clusters of orthologous groups (*Tatusov et al., 2000*). In contrast, the SCOP-domain datasets span all functional classes that can be assigned to homologous sequences. Further, the monophyly of Archaea, and the placement of Archaea sister to Bacteria are supported by the highest PP of 1.0 (Fig. 6). The results are robust to stochastic errors as well as to potential systematic errors related to rate heterogeneity, both CSRH and LSRH. Furthermore, unlike primary sequence data in which compositional bias is a potential source of systematic error, the distinct genomic compositions of unique SCOP-domains are informative about relationships among taxa (*Fang et al., 2013*; *Harish & Kurland, 2017a*). Importantly the use of unique, complex molecular characters, along with directional evolution-models enable the assessment of relationships that extend beyond the phylogeny of a specific group for which suitable outgroups are unavailable. It is also useful in cases where the choice of outgroup sequences is restricted and/or prone to artifacts (e.g., LBA and compositional bias), which can not only confound the placement of the root, but can also influence the ingroup phylogeny (*Graham, Olmstead & Barrett, 2002*).

Moreover, systematic errors in phylogenetic inference (e.g., LBA or model misspecification) are primarily errors in adequately distinguishing homologous similarities

from homoplastic similarities (*Avise & Robinson, 2008*; *Morrison, 2006*; *Philippe et al., 2011*). Homologies, synapomorphies and homoplasies are qualitative inferences, yet are inherently statistical. The probabilistic framework has proven to be powerful for testing alternative hypotheses. Log odds ratios, such as LLR and LBF, are measures of how one changes belief in a hypothesis in light of new evidence (*Huelsenbeck, Larget & Alfaro, 2004*). Accordingly, directional evolution-models are the most optimal explanations of the observed distribution of genomic signatures (Figs. 3 and 6). Such directional trends overwhelmingly support the monophyly of the Archaea, as well as the sisterhood of the Archaea and the Bacteria, that is, monophyly of Akarya as well as monophyly of Eukarya (Fig. 6).

These findings are in stark contrast compared to those of MSA-based analyses (*Nelson-Sathi et al., 2015*; *Rochette, Brochier-Armanet & Gouy, 2014*; *Thiergart et al., 2012*; *Zaremba-Niedzwiedzka et al., 2017*). Since these contrasting results cannot be reconciled, it is worthwhile to revisit the source data that support the conflicting hypotheses. As mentioned earlier, the DDNs derived from single-gene and core-genes data (Fig. 1) as well as from protein-domain data (Fig. 2) are both quantitatively and qualitatively different representations of genomes (Fig. 3). Accordingly, models that describe qualitatively different processes of molecular evolution are required to explain the data. The relevant evolutionary processes/events are mutually exclusive: while the former is explained by point mutations within the selected loci (Fig. 1), the latter is explained by gain and loss (or birth and death) of the selected loci (Fig. 2). Likewise, the sources of the observed conflicts in the DDNs are qualitatively different as well. The sources of the observed conflicts, though, are unknown *a priori* in both cases (Figs. 1 and 2).

In primary sequence data (Fig. 1) conflicts could arise due to stochastic errors (e.g., gene/site sampling and alignment errors) and evolutionary processes (e.g., mutational bias due to uneven damage and repair of genes). Conflicts in the protein-domain datasets (Fig. 2) could arise due to genome/locus sampling errors and evolutionary processes such as ILS or some type of gene flow, including HT. A naïve interpretation of both the DDNs (Figs. 1 and 2) that discounts the complexity of the conflicts would be to assume that all the reticulations represent HT events. Likewise, another naïve interpretation of the marker genes-DDNs (Fig. 1) that disregards the evolutionary polarity is—the Archaea appear to be a chimeric group derived from a fusion of bacterial and eukaryote lineages, and that the members of the group diverged following a fusion event (Figs. 1A–1D).

It is relatively straightforward to distinguish evolutionary signal from noise as per standard phylogenetic theory, provided that the polarity of character transitions can be determined. It is more so for unique phylogenetic characters such as protein-domains, as described earlier. However, it is non-trivial to distinguish between random noise and phylogenetic noise (homoplasy) on the one hand, and between the different causes of phylogenetic noise (*Avise & Robinson, 2008*; *Morrison, 2009*). Existing methods do not distinguish between the different types of noise and hence it is hardly quantified as such. Therefore, there is a tendency to interpret the observed conflicts as evolutionary events (i.e., prior belief), most often as HT events. This is especially the case when

inferences are drawn from analyses of single-gene MSAs (*Murray et al., 2016*).
As emphasized earlier, unrooted trees as such do not distinguish between phylogenetic
signal and noise, let alone distinguishing between the different types of noise. As a
result, identification of "close relatives" or HT based on unrooted trees and other
measures of overall similarities in single-gene MSAs may not be optimal, especially when
the signal to noise ratio is low in a given MSA (*Eisen, 2000*; *Salzberg, 2017*).

Inference of historical HT events is, by necessity, statistical as is any other unobservable
event from the evolutionary past (*Salzberg et al., 2001*). Statistical inferences are as such
robust when a large number of features can be compared. As far as the global ToL is
concerned, MSAs of individual genes are not sufficiently large on their own given the
data quality (Figs. 1 and 3). Conflicting tree topologies are, by and large, associated with
rate- and compositional heterogeneities in both concatenated and single-gene analyses
(*Gouy, Baurain & Philippe, 2015*; *Williams & Embley, 2014*). Several models that
account for (correct for) errors in measuring rate- and compositional heterogeneities in
MSAs have been developed (*Arenas, 2015*; *Gouy, Baurain & Philippe, 2015*). However,
conflicts in tree topologies that arise due to other systematic errors, such as the prevalent
assumptions of reversibility and stationarity of the evolutionary process, are rarely
acknowledged. This is highly relevant, and crucial for rooting trees and for inferring gene
origin and HT events, especially at the grand scale of the ToL.

Indeed, careful measurements of HT show a strong correlation between potential HT
events and systematic error in MSA-based estimates. Most potential HT events inferred
from anomalous placements of gene-OTUs are associated with systematic error, even
among closely related lineages—for example, within a single genus (*Murray et al., 2016*).
These findings suggest that reversible substitution models and unrooted gene trees are
suboptimal tools to diagnose "gains" of genomic loci by HT. Estimation of gains is better
suited for methods that are designed to systematically model gains and losses in genomes
(*Klopfstein, Vilhelmsen & Ronquist, 2015*; *Zamani-Dahaj et al., 2016*). In general,
systematic models of gene (or domain) gain-and-loss estimate significantly lower
frequencies of HT (*Zamani-Dahaj et al., 2016*), compared to HT estimates based on
overall similarities (*Nelson-Sathi et al., 2015*; *Roettger, Martin & Dagan, 2009*).
HT estimates are consistently lower across the ToL: within *Rickettsia*, a genus
(*Murray et al., 2016*); within Cyanobacteria, a phylum and within Archaea (*Zamani-Dahaj
et al., 2016*). In addition, and importantly, these studies show that the tree-like pattern
of inheritance of genomic loci is explained largely by the variation in rates of loss
among lineages, and that the fraction of loci that are prone to HT is a minority
(*Harish, Tunlid & Kurland, 2013*; *Zamani-Dahaj et al., 2016*).

These findings are incompatible with the conventional view that extensive historical
HT has resulted in mosaic genomes in extant species of Archaea (*Nelson-Sathi et al., 2015*),
Bacteria (*Lake, Jain & Rivera, 1999*; *Martin, 1999*), and eukaryotes (*Rochette,
Brochier-Armanet & Gouy, 2014*; *Thiergart et al., 2012*). This incongruence is unsurprising
for the simple reason that these incompatible inferences are drawn from qualitatively
different evolution-models that describe mutually exclusive processes of character
evolution. It will be useful to recall that substitution mutations in genomic loci (Fig. 1) and

gain-loss (birth-death) of loci (Fig. 2) are mutually exclusive. These issues are discussed extensively elsewhere (*Harish & Kurland, 2017a*, *2017c*; *Murray et al., 2016*; *Zamani-Dahaj et al., 2016*).

Sophisticated statistical tests for evaluating tree robustness, and for selecting character evolution-models are becoming a standard feature of phylogenetic software. However, tests for character evaluation are not common even though data quality is at least as important as the evolution-models that are posited to explain the data. Routines for collecting and curating data upstream of phylogenetic analyses are rather eclectic. Besides, it is an open question as to whether qualitatively different datasets (as in Figs. 1 and 2) can be compared effectively. Nevertheless, employing DDNs and other tools of exploratory data analysis would be valuable to identify conflicts that arise due to data collection and/or curation errors. In addition, it is important to recognize the difference between DDNs (undirected networks) and evolutionary networks (directed networks that represent evolutionary history)—just as it is important to distinguish an unrooted tree from a rooted tree (*Morrison, 2006*, *2009*), to draw evolutionary inferences (Fig. 5).

## Additional thoughts on rooting the ToL

Phylogenetic theory as well as related methods of discrete character analysis that were developed for the systematic classification of organismal families (*Darwin, 1859*; *Hennig, 1965*), was embraced, although not entirely, to determine the evolution and classification of gene families (*Woese & Fox, 1977*; *Zuckerkandl & Pauling, 1965*). The initial recognition of the Archaea was based on the comparative analysis of a single-gene (rRNA) family. It is remarkable that the uniqueness of the Archaea was identified by the comparative analyses of oligonucleotide signatures in a single-gene dataset (*Woese & Fox, 1977*). However, the same is not true of the phylogenetic classification of Archaea, based on marker genes and reversible evolution-models. In spite of the large number of characters that can be analyzed, neither the rRNA genes nor multi-gene concatenations of core-genes have proven to be efficient markers to reliably resolve the phylogenetic affinities of the Archaea (*Gribaldo et al., 2010*; *Gupta, 2016*). Consequently, there is a growing consensus that genomes as OTUs (Fig. 2), rather than genes as OTUs (Fig. 1), are not only more informative but are also more appropriate for organizing biodiversity, and for understanding the evolutionary history of species (*Boussau & Daubin, 2010*; *Coenye et al., 2005*; *Harish & Kurland, 2017a*).

Standard evolution-models implemented for phylogenomic analyses are limited to modeling variation in patterns of point mutations. These evolution-models are intimately linked to highly idealized concepts of molecular evolution, such as the universal molecular clock (*Zuckerkandl & Pauling, 1965*), the universal chronometer (*Woese, 1987*), paralogous outgroup rooting (*Schwartz & Dayhoff, 1978*), etc., which are gene-centric concepts that were developed to study the gene, during the age of the gene. Moreover, these idealized notions originated from the analyses of relatively small single-gene datasets. Conventional phylogenomics of multi-locus datasets is a direct extension of the concepts and methods developed for single-locus datasets (*Philippe et al., 2011*). These methods rely exclusively on substitution mutations, which may not be ideal phylogenetic markers

(*Rokas & Holland, 2000*). In contrast, the fundamental concepts of phylogenetic theory: homology, synapomorphy, homoplasy, character polarity, etc., even if idealized, are more generally applicable. And, apparently they are better suited for unique and complex molecular characters rather than for redundant, elementary sequence characters; with regards to determining both qualitative as well as statistical consistency of the data and the underlying assumptions.

In the absence of prior knowledge of outgroups or of fossils, rooting the global ToL is arguably one of the most difficult phylogenetic problems. The conventional practice of *a posteriori* rooting, wherein an unrooted tree is converted into a rooted tree by adding an *ad hoc* root, encourages a subjective interpretation of the ToL. For example, the so-called bacterial rooting of the ToL (root R1; Fig. 5) is the preferred rooting hypothesis to interpret the ToL even though that rooting is not well supported (*Gouy, Baurain & Philippe, 2015*). Incorrect rooting may lead to profoundly misleading conclusions about evolutionary scenarios and phylogenetic relationships, and it appears to be common in phylogenetic studies (*Graham, Olmstead & Barrett, 2002*). For example, root placement between eukaryotes and akaryotes is incompatible with the chimeric origins of eukaryotes (*Harish & Kurland, 2017c*; *Harish, Tunlid & Kurland, 2013*; *Kurland & Andersson, 2000*).

Likewise, because of the central role of phylogenetic inference in biological classification, incorrect rooting or accommodating a priori scenarios (e.g., endosymbiosis or fusion scenarios for eukaryote origins) could confound systematic classification (*Gribaldo & Brochier-Armanet, 2012*); for example, proposals for primary kingdoms (*Whittaker, 1969*; *Woese & Fox, 1977*), primary Domains/Empires (*Harish & Kurland, 2017a*; *Harish, Tunlid & Kurland, 2013*; *Lake, 1986*; *Mayr, 1998*; *Williams et al., 2013*; *Woese, Kandler & Wheelis, 1990*) and other recent proposals for systematic ranks such as Superphylum (*Fuerst, 2013*; *Guy & Ettema, 2011*). Genomic signatures and phylogenetic models that assess the polarity of evolutionary transitions will be valuable to resolve conflicting proposals.

## CONCLUSIONS AND FUTURE DIRECTIONS

The three-domains of life hypothesis (*Woese, Kandler & Wheelis, 1990*) was initially based on the interpretation of an unrooted rRNA tree (of life) (*Woese, 1987*; *Woese & Fox, 1977*). It was put forward largely to emphasize the uniqueness of the Archaea, ascribed to an exclusive lineal descent. The robust support for monophyly of the Archaea based on phylogenetic analysis of genomic signatures agrees with other lines of evidence, molecular, or otherwise (*Garrett, 1985*; *Valentine, 2007*). Idiosyncratic features that support the uniqueness of the Archaea include the subunit composition of supramolecular complexes like the ribosome, DNA- and RNA-polymerases, biochemical composition of cell membranes, cell walls, and physiological adaptations to energy-starved environments, among other things. However, phylogenetic models of the evolution of genomic signatures support a two-domains, or rather two-empires of Life hypothesis (*Mayr, 1998*). Neither the alternative two-domains/Eocyte hypothesis (*Lake, 1986*) nor the three-domains hypothesis (*Woese, Kandler & Wheelis, 1990*)

is supported. Accordingly, genomic evolutionary signatures do not support the presumed primitive state of Archaea and Bacteria (akaryotes), and the traditional belief that Archaea and Bacteria should be ancestors of Eukarya (*Sagan, 1967*; *Spang et al., 2015*; *Williams et al., 2013*; *Woese & Fox, 1977*; *Zaremba-Niedzwiedzka et al., 2017*). The independent origins and parallel descent of eukaryote and akaryote species (*Gouy, Baurain & Philippe, 2015*; *Harish, Tunlid & Kurland, 2013*) is the best-supported hypothesis.

This study shows that phylogenetic inference based on functional genomic signatures and directional evolution-models is less prone to systematic errors due to LBA, CSRH, LSRH, and compositional biases that often mislead MSA-based inferences. Consequently uncertainties in resolving the branches of the ToL, especially the early divergences, can be minimized effectively. The shortcomings of MSAs and substitution models can be overcome by employing complex molecular characters, which initially were thought to be a complementary set of phylogenetic markers that are useful for resolving difficult systematic problems (*Rokas & Holland, 2000*). However, given the qualitative differences of the data types, should MSA-based phylogenetic inferences be supplemented with complex molecular characters and corresponding character evolution-models? Or perhaps supplanted? I argue for the latter based on the findings of this study, and the limited perspective that is provided by the core-genes datasets toward understanding the early diversification of the ToL. The resolving power of gene-sequences using substitution models has been overstated—if not in general, it is evidently the case with regards to resolving the early diversification of Archaea and the placement of the root of the global ToL. Employing genomic signatures is particularly relevant to study the evolution of the biodiversity of uncultivable microbial species that is characterized by genome sequences.

It is worth emphasizing that the impact of LSRH (heterotachy) was not assessed in almost all recent studies that characterized incongruences in various phylogenomic datasets, including those of core-genes datasets. It appears that accounting for LSRH is unlikely to improve the analyses of core-genes datasets, though, it is a potential source of systematic error for the larger datasets such as those used to resolve the root of the metazoan-ToL. Perhaps, a stronger potential for systematic error is the assumption of reversibility and stationarity in standard evolution-models. Both assumptions are made for mathematical simplicity and computational convenience, but may not be biologically realistic (*Kaehler, Yap & Huttley, 2017*; *Morrison, 2006*). Computational limitation is a major factor for implementing directional evolution-models for large datasets that employ multi-state characters including MSA datasets. Regardless, exclusive reliance on a single data type, and a single evolutionary process (i.e., substitution mutation) might not be sufficient for resolving all phylogenetic relationships. Historical signals in MSAs and other data types relate to qualitatively different, and mutually exclusive evolutionary processes that cannot be modeled simultaneously. Therefore, polyphasic analyses, rather than a combined analysis of different data types that are informative at different phylogenetic depths could be useful.

## ACKNOWLEDGEMENTS

I am grateful, foremost, to David Morrison and Charles Kurland for stimulating discussions. David Morrison, Charles Kurland, Måns Ehrenberg, and Suparna Sanyal for support and encouragement. Siv Andersson for inspiring the article title (in part) and general discussion. Seraina Klopfstein for providing the algorithms for implementing the directional model in MrBayes, and for helpful suggestions. David Morrison, David Pollock, David Polly, and Kenneth Halanych for comments on an earlier version of the manuscript as well as Bruce Lieberman, and Joseph Gillespie for thoughtful comments that helped improve the presentation; two anonymous reviewers for critique. Thijs Ettema for providing the SSU rRNA and Core-genes-II MSAs, and Erling Wikman for help with computing equipment.

### Funding

This research received no specific grant from any funding agency in the public, commercial, or not-for-profit sectors. Work by this author was partially supported by the Swedish Research Council (to Måns Ehrenberg) and the Knut and Alice Wallenberg Foundation, RiboCORE (to Måns Ehrenberg and Dan Andersson). Article processing charge was supported by research grants from the Swedish Research Council, Research Environment Grant dnr: 2016-06264 and Knut and Alice Wallenberg Foundation, KAW 2017.0055 (to Suparna Sanyal). The funders had no role in study design, data collection and analysis, decision to publish, or preparation of the manuscript.

### Grant Disclosures

The following grant information was disclosed by the authors:
The Swedish Research Council.
The Knut and Alice Wallenberg Foundation, RiboCORE.
The Swedish Research Council, Research Environment: 2016-06264.
Knut and Alice Wallenberg Foundation: KAW 2017.0055.

### Competing Interests

The authors declare that they have no competing interests.

### Author Contributions

- Ajith Harish conceived and designed the experiments, performed the experiments, analyzed the data, contributed reagents/materials/analysis tools, prepared figures and/or tables, authored or reviewed drafts of the paper, approved the final draft.

### Data Availability

The raw data (character matrices) are provided as nexus files in the Supplemental Information.

## Supplemental Information

Supplemental information for this article can be found online at http://dx.doi.org/10.7717/peerj.5770#supplemental-information.

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
