# Peer review of "What is an archaeon and are the Archaea really unique?"

_PeerJ, doi:10.7717/peerj.5770_

## Round 0.1 · original submission · Major Revisions

Dear Dr. Harish:

Thanks for submitting your work to PeerJ. I apologize for the lengthy review process. You provided many potential reviewers, which helped, but because of the contentious nature of your manuscript, I do feel many appropriate scientists refused to review your work. Nonetheless, I have finally received three independent reviews of your work, and as you will see, the reviewers raised some concerns about the research. Per reviewer 1, I agree that the section “Sorting Vertical Evolution (Signal) from Horizontal Evolution (Noise)” should be renamed to avoid confusion (I do think that the reviewer’s suggestion “Sorting Evolutionary Signal from Noise”) is a better fit. I also agree with the reviewer that you must open the dialogue on lateral gene transfer as it applies to not only the creation of these major lineages, but also to phylogeny estimation. This issue is obviously important for any phylogenetic investigation, let alone one with far-reaching implications.
I also agree with reviewer 1 that the section “Vertical and Horizontal Classification” begins with a dialogue that is in many aspects outdated and possibly irrelevant to your work. Reviewer 2 also struggled with this entire section. Please revise accordingly. Also, this section is not aptly named, mostly because the terms “vertical” and “horizontal” apply to different aspects of phylogeny estimation, but tied to classification makes them a bit odd. The suggestion by the reviewer (“Additional Thoughts on Rooting the TOL”) is simpler and more effective. I strongly recommend changing this, and also keeping in mind that simpler language throughout the manuscript will make the work more effective and certainly less ambiguous.
Reviewer 3 raises a very important issue with regards to the chimeric nature of eukaryotic genomes. Please remember to address the widely-acknowledged origin of Eukaryota from archaeal and bacterial genomes. I am stunned that the seminal work by Bill Martin and colleagues (“An Evolutionary Network of Genes Present in the Eukaryote Common Ancestor Polls Genomes on Eukaryotic and Mitochondrial Origin”, PMID: 22355196) is not discussed, or related articles. The reviewer raises an excellent argument about protein domain evolution, and how your rooting approach fails to account for the chimeric nature of eukaryotic genomes. This must be addressed in your revision. Please consider the analyses of your dataset that the reviewer performed. It would be ideal if you could include such an analysis in your revisions that could directly account for the chimeric nature of eukaryotes. The rooting and basal diversification seem to be quite a bit sensitive to a variety of approaches, so the more permutations of the data (and methods) the better.
Please thoroughly assess all of the additional points raised by all of the reviewers…they all need to be addressed in your revision.
Minor points: I agree with reviewer 2 that something seems missing from the dialogue on lines 663-667. Did text get deleted? Otherwise rephrase the English in this section. I disagree with reviewer 3 regarding the length of your manuscript. In any case, if you so see ways to trim it down in your revision, please do so (if some aspects may be moved to a supplemental document, feel free to do so to shorten the dialogue). In general, please address the grammatical suggestions raised by the reviewers, as well as missing references. Your paper is well-written and researched, but still can be improved in this regard.
Your work is obviously of great interest to a broad community of evolutionary biologists, and I encourage you to proceed with this revision. I look forward to seeing your next effort, and thanks again for submitting your work to PeerJ.

Good luck with your revision,

-joe

·

Basic reporting

This paper is somewhat out of my direct area of expertise, though I consider it a topic of broad interest that is likely to generate very wide readership. It is well written and takes on an issue that has much relevance for understanding phylogenetic patterns in the tree of life at the grand scale. I particularly liked how the author revisited some of the core issues and debates about the potential problems with a 3 branched tree of life that were originally brought up by Jim Lake. The 3 branched view of the tree of life stood in constradistinction to the 2 branched tree of life (TOL) that recognized the “Eocytes” (that Lake propounded). The result and conclusions the author puts forward are likely to be somewhat controversial (I’m not saying they’re wrong, just that others may disagree), but that is okay. I think the paper will stimulate discussion and it represents a useful addition to the larger debate about the existence of Archaea and major patterns in the TOL, the data and methods supporting the grouping, etc.

There are a couple of sections, towards the end of the paper, for which I might recommend some changes. Nothing major, but I think these might tighten up and better focus the paper. First, in the “Sorting Vertical Evolution (Signal) from Horizontal Evolution (Noise)” section. Initially, with that title, I thought the author was going to mention horizontal gene transfer. Perhaps here it might be worth mentioning what impact, if any, that phenomenon could have on phylogenetic relationships. Also, I think I’d just change the title of this section to “Sorting Evolutionary Signal from Noise”.

Then, for the next section, I found the discussion of “Vertical and Horizontal Classification” a bit confusing, partly because those are terms that I am not familiar with, at least in this context. I’d recommend changing the title of this section to: “Additional Thoughts on Rooting the TOL”. Then the Simpson (1945) reference is very out of date, and I don’t think most readers will be familiar with it, nor the vertical and horizontal terminology that comes from there. I’m also not sure if phylogenetically minded folks will agree with all of the statements pertaining to the first paragraph of that section, such that it could detract from the overall message of the paper. Thus, I’d also recommend deleting the first paragraph in that section. Instead, just begin with “The classical rooting of the … “.

In the first paragraph of the Conclusions section, I’d cite Lake (1986) instead of, or in addition to, Mayr (1988).

Also, a couple of very minor comments:
Add the word “to” between “Due” and “a” on line 172.
Line 432, change to- Depend on “the” notion

Experimental design

All looked good to me, no other comment.

Validity of the findings

All looked good to me, no other comment.

Additional comments

An excellent publication that I think will make a very worthy addition to PeerJ.

Reviewer 2 ·

Basic reporting

The English is good and overall the paper is well-written. I see some mistakes in English in a few places, but nothing stood out terribly.


Introduction, Methods and Result are quite clear and detailed. I have more concerned with the discussion where even if I see myself agreeing with some parts (especially with the importance of the quality of the data), I am not quite sure to have fully understand some others, especially "Vertical and Horizontal Classification" and "Untangling Data Bias, Model Bias and Investigator Bias (Prior Beliefs)" (see general comments for additionnal comments).

Experimental design

no comment

Validity of the findings

no comment

Additional comments

Abstract: I have some questions mark on my end (line : I find that the uncertainty is primarily due to a
scarcity of information in standard datasets ? universal core genes datasets ? to reliably resolve the
conflicts.). Is this normal ? If it is, I am not quite sure to understand this part of the abstract.

663-667: I have trouble understanding these part. I'm under the impression something is missing there so I would rephrase it.

Reviewer 3 ·

Basic reporting

The paper is fairly clear, though a long-winded and repetitive in places. I found the analysis of the alternative rootings confusing (root-R1 etc). It is also unclear whether this paper represents much of an advance over prior work by the same author, who seems to have been plugging this basic outlook since at least 2013. The paper would be much improved by being halved in length.

Experimental design

On the positive I do believe that the idea of using directional characters to root the tree of life is an important and worthwhile insight, and agree that conserved protein domains have built in directionality, being hard to gain in parallel but easy to be lost in parallel. However, there is a major problem with the approach that the author does not seem to acknowledge: eukaryotes are pretty clear chimeric representing a merger of both an archaeal and bacterial proteome. This is quite devastating in that this predicts that eukaryotes will have a greater diversity of protein domains than either bacteria or archaea, which will tend to pull eukaryotes to the root of the tree (because any parallel gains can be "saved" by placing them on the root lineage). I would consider the paper to be unpublishable unless the author can find a way to properly handle the chimeric nature of eukaryotes.

In order to overcome this issue it night be possible to split eukaryotes into 2 pseudo taxa, one containing domains that probably came from proteobacteria (or cyanobacteria in the case of archaeplastida) and one containing domains of archaeal (Asgardian?) origins. However, without some such correction I cannot see how this work could help support the three-domains topology. Indeed, when I reanalyzed the larger data set using asymmetric step matrices and a parsimony criterion after deleting all eukaryotes (a pain since they were not clearly grouped in the dataset), the resulting trees were rooted with a paraphyletic bacteria, which is incompatible with a three-domains topology regardless of where eukaryotes fit. (Also, when I created and chimeric taxon that was about half bacterial and half archaeal it dropped down towards root.)

The author draws a distinction between CSRH models with different number of categories used to discretize the gamma distribution. It should be noted that these all entail the same number of parameters and that, empirically, there is rarely any effect of going above 4 rate categories.

The author notes: "...complex features — including molecular, biochemical and phenotypic characters, as well as ecological adaptations —support the uniqueness of the Archaea." However, this is misleading in that these traits could all be plesiomorphic for an archaeal+eukaryotic clade, with subsequent modification in eukaryotes.

I do not find the splitstree analyses useful and recommend that they be dropped.

Validity of the findings

I believe that the analyses are done well but that the fundamental flaw in the experimental design invalidates the conclusions.

Additional comments

I am not especially committed to any one rooting of the universal tree, so my criticism here is not based on any allegiance to the eocyte tree beyond my sense that it has support that is not easy to discount as being due to a particular driving artifact.

I will check "major revision" but you should be aware that unless you can effectively deal with the core problem of eukaryotic chimerism, this might as well be considered a "reject."

---

## Round 0.2 · Minor Revisions

Dear Dr. Harish:

Thanks for re-submitting your manuscript to PeerJ. I have now received two reviews of your revision, and as you will see, both are favorable, though one reviewer (original reviewer 3) still has some issues. I would like you to address these concerns in order for us to move towards publishing your work. I agree with reviewer 3’s concerns, and I’m sure you know that your stance on eukaryote evolution is a bit different than the consensus of the field. This is fine, and frankly invigorating for the dialogue, but I do think the suggestions of reviewer 3 will make for a better manuscript, one that will draw a broader interest of the community.

Therefore, I am recommending that you revise your manuscript accordingly, taking into account the issues raised by reviewer 3. I do believe that your manuscript will be ready for publication once these issues are addressed.

Good luck with your revision,

-joe

·

Basic reporting

The author has made all of the changes I've requested, and it looks like he's done a good job addressing the changes that others have requested as well. Thus in my view the paper is ready to move on to the next phase of production and be published.

Experimental design

No comment

Validity of the findings

The paper is a very strong one in all respects.

Additional comments

Nice job addressing my comments.

Reviewer 3 ·

Basic reporting

The basic reporting expectations are met by this paper. Nonetheless, I continue to believe that the paper would be much better received (and more likely to be cited) if it did not attempt to lay out every argument for the author's pet theory, all of which he has already published, and instead stuck to what is new. The narrative structure would be simple. "As I have showed previously [refs], the presence/absence of protein domains provides a robust method for the inference of rooted trees. Thus, provided one accepts that there has been no major chimerism - i.e., that the endosymbiotic theory for the origin of eukaryotes is incorrect [Harish & Kurland 2013] - this approach is well suited to resolving the overall structure of the rooted tree of life. Here I analyze a larger/better data set of domains and, using improved methods, re-evaluate my prior conclusion that eukaryotes are sister to "akaryotes" (=bacteria+archaea). I find...."

It is very important that the reader be told in the introduction that the entire method assumes that mitochondria arose autogenously. It is an attempt to better root the tree of life, under this (controversial) assumption.

Experimental design

I stand by my prior concerns regarding chimerism. First, to clarify I used a 10:1 step matrix and did indeed find a root within bacteria (I just checked). Second, it all turns on the author's MAJOR assumption that the endosymbiotic hypothesis is wrong. I went back an looked at the key paper, Harish and Kurland (2013), and am not at all convinced! Let me provide a quick critique.

The paper makes the major mistake of equating the mitochondrial proteome with the endosymbiont. To be sure we would expect the mitoproteome to be ENRICHED for endosymbiotic genes, but we need to remember that under the endosymbiotic hypothesis, (a) very many bacterial genes were transferred to the nucleus and acquired distinct functions outside of the mitochondrion, and (b) some proteins of archaeal ancestry would have acquired mitochondrial target sequences and mitochondrial functions. With than in mind lets look at their data, bearing mind that for some reason Archaea seem to have fewer domains than the other groups (ignoring those shared by all groups, the numbers are: Arch = 156; Bact = 419; Euk = 738): In the mitoproteome, seven times as many genes are uniquely shared with bacteria (and not archaea) than the reverse. This contrasts with only 2.5X enrichment for the non-mitochondrial proteome. To my mind this over-enrichment of bacterial genes in the mitoproteome is fully consistent with the endosymbiotic model!

I would also ask the author to explain away the many gene trees with robust support for eukaryotic genes that originated from within the proteobacteria - including some where the analysis used non-stationary models as the author favors (e.g., Wang and Wu, 2015).

Validity of the findings

At the end of the day I believe the author is wrong in his rejection of the endosymbiotic model. This invalidates the method as it stands.

Additional comments

There are three choices:
a) Push ahead with this method and present a result that will be completely ignored by anybody who shares my view - and that is pretty much all biologists.
b) Accept that the endosymbiotic model could be correct, in which case the method could be used to evaluate competing models for where eukaryotes sit and where the root goes. Such an analysis would be of broad interest. You would need to split the eukaryotes into two taxa, for example one taxon being composed only of domains found anywhere in bacteria and anywhere in eukarya, plus a random half of the eukaryote-specific domains, and the other being composed only of domains found anywhere in archaea and anywhere in eukarya, plus the other half of the eukaryote-specific domains. But there may be other strategies.
c) Do both in the same paper.

I recommend option b or, failing that, c.

---

## Round 0.3 · accepted · Accept

Dear Dr. Harish:

Thanks for addressing the concerns raised in the revision. I do believe that your work is now suitable for publication. I do hope that it generates a healthy and stimulating dialogue on the evolution of life's major lineages. Congratulations!

Thanks for submitting to PeerJ!

Best,

-joe

#